# A versatile functionalized ionic liquid to boost the solution-mediated performances of lithium-oxygen batteries

Jinqiang Zhang [1], Bing Sun[1], Yufei Zhao[1,2], Anastasia Tkacheva[1], Zhenjie Liu[3], Kang Yan[1], Xin Guo[1], Andrew M. McDonagh[1], Devaraj Shanmukaraj[4], Chengyin Wang[5], Teofilo Rojo[4], Michel Armand[4], Zhangquan Peng[3] & Guoxiu Wang [1]

Due to the high theoretical specific energy, the lithium–oxygen battery has been heralded as a promising energy storage system for applications such as electric vehicles. However, its large over-potentials during discharge–charge cycling lead to the formation of side-products, and short cycle life. Herein, we report an ionic liquid bearing the redox active 2,2,6,6-tetramethyl-1-piperidinyloxy moiety, which serves multiple functions as redox mediator, oxygen shuttle, lithium anode protector, as well as electrolyte solvent. The additive contributes a 33-fold increase of the discharge capacity in comparison to a pure ether-based electrolyte and lowers the over-potential to an exceptionally low value of 0.9 V. Meanwhile, its molecule facilitates smooth lithium plating/stripping, and promotes the formation of a stable solid electrolyte interface to suppress side-reactions. Moreover, the proportion of ionic liquid in the electrolyte influences the reaction mechanism, and a high proportion leads to the formation of amorphous lithium peroxide and a long cycling life (> 200 cycles). In particular, it enables an outstanding electrochemical performance when operated in air.

[1] Centre for Clean Energy Technology, University of Technology Sydney, Broadway, Sydney, NSW 2007, Australia. [2] Department of Materials Science and Engineering, Dongguan University of Technology, Dongguan, Guangdong 523808, People's Republic of China. [3] State Key Laboratory of Electroanalytical Chemistry, Changchun Institute of Applied Chemistry, Chinese Academy of Sciences, Changchun, Jilin 130022, China. [4] CIC EnergiGUNE, Parque Tecnológico de Álava, 48, 01510 Miñano, Álava, Spain. [5] College of Chemistry and Chemical Engineering, Yangzhou University, Jiangsu 225002, People's Republic of China. Correspondence and requests for materials should be addressed to M.A. (email: marmand@cicenergigune.com) or to Z.P. (email: zqpeng@ciac.ac.cn) or to G.W. (email: Guoxiu.Wang@uts.edu.au)

Lithium oxygen (Li–O₂) batteries possess the highest theoretical energy density among all rechargeable batteries[1–4]. Typically, a Li–O₂ cell consists of a lithium metal anode, a porous cathode, and a separator saturated with electrolyte[5]. Oxygen can be drawn directly from the ambient atmosphere during discharge to form the discharge product of lithium peroxide ($Li_2O_2$). The reaction can be reversed during the charging process. However, due to its insulating nature, $Li_2O_2$ deposited on the cathode during discharge passivates the surface of cathode, resulting in the formation of large amount of unwanted side-products such as $Li_2CO_3$[6–8]. This leads to a low reversible capacity and poor cycle life of Li–O₂ batteries. The electrochemically formed $Li_2O_2$ usually has high crystallinity. The decomposition of such crystalline $Li_2O_2$ during charge process requires additional energy input, leading to an increase of charge potentials, which further causes side-reactions. These drawbacks significantly inhibit the development of high performance Li–O₂ batteries.

Various catalysts have been employed to facilitate the formation and decomposition of $Li_2O_2$, thereby increasing the efficiency of Li–O₂ batteries[9–17]. However, catalysts often require direct contact between the catalysts and $Li_2O_2$ particles. The lack of sufficient particle-to-particle contacts reduces round-trip efficiencies and results in short cycle life[18]. Solution-based mediators, on the other hand, have been proposed as shuttles within the electrolyte to overcome this problem[15,19]. Oxygen shuttles such as phthalocyanine (PC), 2,5-di-tert-butyl-1,4-benzoquinone (DBDQ), coenzyme Q10, and heme (biomolecule) are reduction mediators that can enhance the solution-phase formation of $Li_2O_2$ in the discharging process by interacting with intermediates including superoxides[20–25]. This reduces the side-reactions originating from the direct attack of superoxide radicals on the solvent molecules, and significantly improves discharge capacities. Redox mediators such as tetrathiafulvalene (TTF), tetramethylpiperidinyloxyl (TEMPO) and lithium halides have been used as electron shuttles to facilitate the decomposition of $Li_2O_2$ during the charge process, creating an alternative pathway for electron transport to improve the charge efficiency, which effectively decreases charge over-potentials[22,26–31]. However, the use of solution-based mediators often causes corrosion of the lithium metal anode[32,33]. Creating a protective layer on the surface of the lithium anode is, therefore, a critical challenge. One approach is to insert separation layers as physical barriers to prevent the direct access of the solution-based mediators to the lithium metal anode[24–38]. For instance, a combination of redox mediator, an oxygen shuttle, and a lithium protection layer can enhance electrochemical performance in Li–O₂ batteries[39]. The protection layers may be relatively thick, which can detrimentally increase the internal resistance of the batteries. To overcome this drawback, a "self-defense" redox mediator, InI₃, was reported to form a lithium protection layer during battery operation instead of adding an external protection layer[40]. Another approach to maintain the integrity of the anode is to constrain the redox mediators to the cathode area. For example, the combination of a redox mediator and a negatively charged surfactant can restrict the movement of the oxidized redox mediator during charge to protect the lithium anode[41]. We have previously shown that oxidized TTF interacts with LiCl to reversibly form an organic conductor, which selectively deposits on the cathode surface during charge to enhance the overall efficiency[42]. Nevertheless, side-reactions are still inevitable when solvents such as dimethyl sulfone (DMSO) and glymes are used[43].

In this work, we incorporate all the advantages of redox mediators and lithium metal protection additive in a multifunctional TEMPO-grafted ionic liquid (IL-TEMPO, Fig. 1a and Supplementary Table 1). The n-/p-doping property of the TEMPO moiety enables IL-TEMPO to function as a redox mediator and an oxygen shuttle, leading to a significantly increased discharge capacity of 33-fold and a dramatically reduced charge voltage of 3.6 V. Furthermore, a stable solid electrolyte interface (SEI) is formed, which ensures smooth lithium stripping and plating. At high concentrations, the IL-TEMPO can efficiently extend the cycle life of the Li–O₂ battery to 200 cycles with the formation of amorphous $Li_2O_2$ as discharge products. Furthermore, the unique properties of IL-TEMPO allow the cell to operate in harsh environment such as at elevated temperature of 70 °C or in an air atmosphere with outstanding electrochemical performances.

## Results

**Redox behavior of IL-TEMPO in DEGDME solution.** The synthesis of the new ionic liquid 1,2-dimethyl-3-(4-(2,2,6, 6-tetramethyl-1-oxyl-4-piperidoxyl)-pentyl)imidazolium bis(trifluoromethane)sulfonimide (IL-TEMPO) is illustrated in Supplementary Fig. 1a[44,45]. Briefly, 2,2,6,6-tetramethyl-4-piperidinol was oxidized using Na₂WO₄ to yield 4-hydroxy-TEMPO followed by reacting with 1,5-dibromopentane to form 5-TEMPO-pentyl bromide. After attachment of 1,2-dimethylimidazole, ion exchange using lithium bis(trifluoromethane)sulfonimide (LiTFSI) yielded IL-TEMPO, which is a red, viscous ionic liquid immiscible with water. IL-TEMPO was characterized by hydrogen nuclear magnetic resonance (¹HNMR, Supplementary Fig. 1b) and Fourier transform infrared spectroscopy (FTIR, Supplementary Fig. 2), to confirm its molecular structure and high purity.

The redox properties of IL-TEMPO were measured using a two-electrode cell with a Swagelok-type configuration (Fig. 1b and Supplementary Fig. 3). Lithium foil, a carbon paper electrode (Supplementary Fig. 4), and a glass fiber membrane were used as anode, cathode, and separator, respectively. The IL-TEMPO electrolyte comprised of IL-TEMPO in diethylene glycol dimethyl ether (DEGDME, the IL-TEMPO volume ratio: 1%) with LiTFSI (0.5 M). Two highly reversible pairs of redox peaks are observed. The peaks at 3.75 V can be assigned to oxidation of the N-O radical in IL-TEMPO, and the peaks at 3.0 V correspond to the reduction of N-O radical in IL-TEMPO (as shown in Fig. 1b)[10,46]. In order to enhance both the discharge and charge processes during the operation of Li–O₂ batteries, the reduction potential of the mediator molecules should be very close to the theoretical formation potential of $Li_2O_2$ and higher than the actual discharge plateau (~2.7 V), while the oxidation potential should be higher than the theoretical decomposition potential of $Li_2O_2$ yet lower than the actual charge plateau (~4.2 V)[20,21]. The potentials of the reversible peaks (3.0 V and 3.75 V) of IL-TEMPO perfectly match the above required potential windows where the non-aqueous oxygen reduction reaction (ORR) and oxygen evolution reaction (OER) are enhanced in Li–O₂ batteries (theoretical potential 2.96 V, Fig. 1b and Supplementary Note 1). Cyclic voltammetry (CV) using an oxygen atmosphere (Supplementary Figs. 5 and 6 and Note 2) shows that the ORR and OER are significantly improved when IL-TEMPO is added to the DEGDME electrolyte. Therefore, the addition of IL-TEMPO could be significantly beneficial for the operation of Li–O₂ batteries during both discharge and charge processes.

**IL-TEMPO as oxygen shuttle.** The electrochemical performances of Li–O₂ cells were evaluated by discharging the cells to 2.0 V followed by reversible charging. The discharge and charge curves shown in Fig. 1c, d are consistent with the aforementioned CV result (Fig. 1b) with a discharge plateau of 2.7 V and a charge plateau of 3.6 V. The over-potential of Li–O₂ cell is significantly reduced from 1.7 V to 0.9 V when IL-TEMPO is added in the electrolyte, demonstrating the enhancement of both discharge

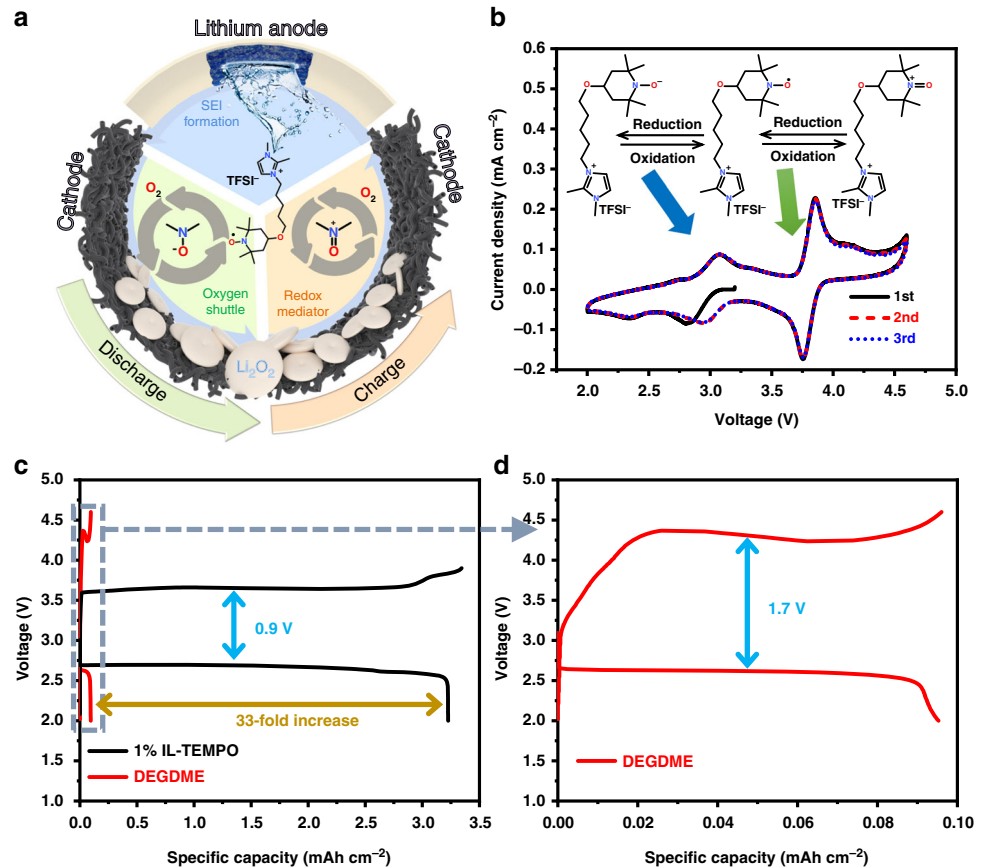

**Fig. 1** Illustration of IL-TEMPO facilitating the performance of Li–O₂ batteries. **a** Schematic illustration of the IL-TEMPO facilitating the performance of Li–O₂ batteries. **b** Cyclic voltammetry curve of battery with IL-TEMPO electrolyte in argon atmosphere. The scan rate is 0.5 mV s⁻¹. The inset image is the illustration of n-/p-doping of IL-TEMPO. The lithium anode is pre-treated with IL-TEMPO-containing propylene carbonate (PC) electrolyte for 5 days before use. **c** The discharge–charge profiles of a Li–O₂ battery with the 1% IL-TEMPO electrolyte and DEGDME electrolyte. **d** The enlarged discharge–charge curves of a Li–O₂ battery with DEGDME electrolyte. The current densities were 0.1 mA cm⁻²

and charge processes with the aid of IL-TEMPO. Surprisingly, the discharge capacity with IL-TEMPO electrolyte is more than 33 times that of the pure DEGDME electrolyte (Fig. 1c, d). The increase of the discharge capacity is dramatic, owing to the exceptional n-doping capacity which may efficiently lower the discharge barrier and enhance the oxygen dissolution[47].

The morphologies of the discharge products are significantly different (Fig. 2a–e). When discharge in a DEGDME electrolyte, the discharge product only consists of a small amount of nanoparticles (Fig. 2b, c). However, a large quantity of particles with toroidal-like morphology was formed when discharged in the IL-TEMPO electrolyte (Fig. 2d, e). The particles have been identified to be $Li_2O_2$ by X-ray diffraction (XRD, Supplementary Fig. 7). The electrode could recover to its original state after charge, indicating the high reversibility (Supplementary Fig. 8). The increase of the discharge capacities is attributed to the exceptional capability of IL-TEMPO to promote the solution formation of $Li_2O_2$ by interacting with the discharge intermediate superoxide radicals ($O_2^{\bullet-}$). There are two possible mechanisms of the interactions between IL-TEMPO and $O_2^{\bullet-}$, depending on which part of the IL-TEMPO is functioning: (i) the reduced form of the TEMPO group, and (ii) the imidazolium cation. Both of them enhance the oxygen dissolution during the discharge process. The TEMPO route shown in Fig. 2f involves two main steps. Oxygen molecules are reduced by the reduced form of the N-O radical, and further interacted with the N-O group through the formation of a lithium bond[48]. The unstable intermediate

rapidly decomposes to form $Li_2O_2$ when it migrates to the surface of the cathode. A second possible route (imidazolium route) involves the direct interaction between the large imidazolium cation and the intermediate $O_2^{\bullet-}$ through electrostatic attraction[49,50]. A demonstration experiment was conducted to simulate the electrochemical process by chemical reduction of the TEMPO group using phenylhydrazine ($PhNHNH_2$), which is commonly used to reduce TEMPO-containing compounds for NMR measurements[51]. The mixture of lithium salt and reduced IL-TEMPO was purged by oxygen for 5 min and casted onto a FTIR attenuated total reflection (ATR) crystal for characterization. We discovered an intriguing phenomenon that oxygen bubbles were generated, whereas there was no additional peaks identified by FTIR (Supplementary Fig. 9, Note 3, and Movie 1). This observation clearly indicates that a highly unstable intermediate involving oxygen species is formed with the assistance of reduced IL-TEMPO to enhance the oxygen dissolution. This could explain the enhancement of discharge capacities (Fig. 1c) and the proposed discharge mechanisms (Fig. 2f).

**IL-TEMPO as redox mediator**. The discharge–charge profiles of Li–O₂ cells with a curtailing capacity are shown in Fig. 3a and Supplementary Fig. 10. The discharge–charge voltages are identical to the full discharge–charge curves in Fig. 1c. Moreover, the Li–O₂ cell shows a good cycle life (Fig. 3a) and rate performance

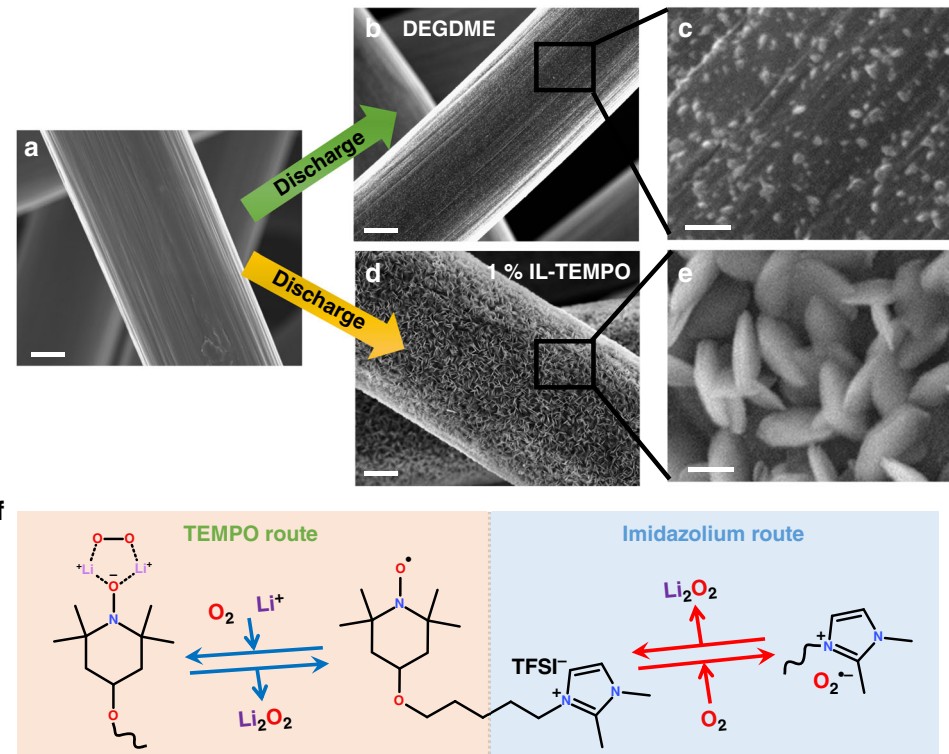

**Fig. 2** The promotion of Li$_2$O$_2$ formation with the aid of IL-TEMPO. **a–e** The scanning electron microscope (SEM) images of the carbon paper electrodes **a** before and **b–e** after the discharge process with **b**, **c** DEGDME electrolyte and **d**, **e** IL-TEMPO electrolyte. (**c** and **e** are the enlarged SEM images of **b** and **d**.) Scale bars are 2 μm in **a**, **b**, **d**, and 200 nm in **c**, **e**. **f**. The mechanism of the discharge facilitation using IL-TEMPO. Both routes involve the formation of intermediates with the oxygen species

(Supplementary Fig. 11), indicating the highly reversible and efficient redox property of IL-TEMPO. To further analyze the redox behavior of IL-TEMPO electrolyte by comparing it with DEGDME electrolyte, carbon nanotube (CNT, Supplementary Fig. 12) electrodes are employed to replace carbon paper electrodes, due to the larger surface area of CNT compared with carbon paper (Supplementary Fig. 13) to better accommodate the discharge products. The comparison of the electrochemical performances is shown in Fig. 3b. The charging plateau voltage of the Li–O$_2$ cell with added IL-TEMPO stabilizes at 3.6–3.7 V (similar to carbon paper), which is ~0.8 V lower than that using a pure DEGDME electrolyte. The dramatic decrease of the overpotential during the charge process is attributed to the reversible redox p-doping of IL-TEMPO, as indicated by CV (Fig. 1b). Differentiation of the discharge–charge curves (Supplementary Fig. 14) shows a similar peak distribution to the CV curve. The reversible capacity observed is not contributed by the self-redox reaction of IL-TEMPO, because the cell operating in an argon atmosphere has a reversible capacity of only 0.03 mAh cm$^{-2}$ (Supplementary Fig. 15), which is negligible compared to the cutoff capacity set for the operation of the Li–O$_2$ cell. Additionally, the extension of the discharge–charge capacity does not result in a second plateau during charge (Supplementary Fig. 16). Thus, the capacity mainly originates from the reversible formation and decomposition of the Li$_2$O$_2$ discharge product (confirmed by XRD, SEM, and FTIR in Supplementary Figs. 17 and 18 and demonstration experiment in Supplementary Fig. 19 and Note 4), which is facilitated by the reversible redox activities of IL-TEMPO. Owing to the high redox reversibility of IL-TEMPO, the charging plateau of each cycle does not change significantly during the continuous operation of the Li–O$_2$ cells. Furthermore, as proven by our designed experiment in Supplementary Fig. 20,

IL-TEMPO is only fully capable of decomposing Li$_2$O$_2$, but not Li$_2$CO$_3$ with high crystallinity. Therefore, there should be a negligible amount of side-products formed (confirmed by titration of Li$_2$O$_2$ in Supplementary Table 2)[52], which leads to the exceptional cycling performance as shown in Supplementary Fig. 21. The Li–O$_2$ cell with IL-TEMPO could easily reach 100 cycles with no visible decay of the capacity, while the one with pure DEGDME electrolyte shows decreased capacity at the 37 cycles. It is worth noting that the extension of discharge/charge capacity does not significantly deteriorate the cycling performance, even with the risk of generating more by-products due to the parasitic reactions between the discharge product Li$_2$O$_2$ and ether-based electrolyte (Supplementary Fig. 22 and Note 5). Furthermore, the IL-TEMPO electrolyte shows exceptional stability, which induces minimum side-reactions during long-time cycling (Supplementary Fig. 23 and Note 6). The rate performance of the IL-TEMPO electrolyte is shown in Supplementary Fig. 24. The current densities do not significantly influence the discharging and charging voltages, owing to the efficient catalytic property of IL-TEMPO.

To identify the chemical reactions during discharge and charge processes, quantitative in situ differential electrochemical mass spectrometry (DEMS) was employed to monitor the gas consumption and evolution during the discharge and charge processes in Li–O$_2$ cells. As shown in Fig. 3c, oxygen is continuously consumed during the discharge process and the ratio of the transferred charge to oxygen consumed (e$^-$/O$_2$) is calculated to be 2.07 (Supplementary Table 3). This verifies that the dominating discharge reaction is the formation of Li$_2$O$_2$ in the DEGDME electrolyte. A similar ratio is also detected in the electrolyte containing IL-TEMPO (Fig. 3e). However, the charging behaviors differ considerably when the different electrolytes are used. In Fig. 3d, the DEGDME electrolyte

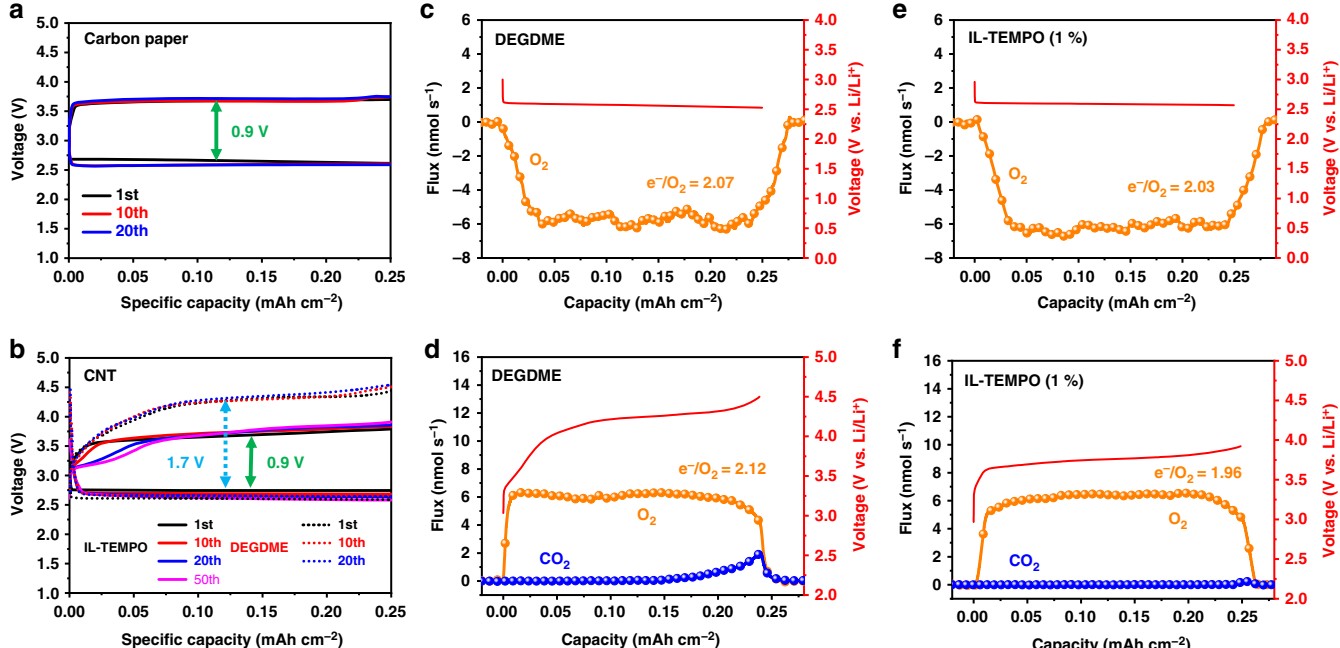

**Fig. 3** Electrochemical performances with fixed capacities. **a**, **b** The discharge–charge profiles of a Li–$O_2$ battery with **a** carbon paper electrodes and **b** carbon nanotube (CNT) electrodes. The current densities were 0.1 mA $cm^{-2}$, and the cutoff voltage was 2.3 V/4.6 V. **c–f** In situ differential electrochemical mass spectrometry (DEMS) analysis of the gas consumption and evolution during Li–$O_2$ cell operation: **c**, **e** discharge to 0.25 mAh $cm^{-2}$ at current density of 0.25 mA $cm^{-2}$, and **d**, **f** charge to 0.25 mAh $cm^{-2}$ at current density of 0.125 mA $cm^{-2}$. The error of the DEMS data obtained during discharge is 5%, and during charge is 3%

results in a charging plateau as high as 4.2 V, which is consistent with Fig. 1c. At the end of the charging stage, a significant amount of $CO_2$ is detected, indicating the decomposition of the electrolyte solvent and side-products accumulate during discharge and charge processes. In contrast, the electrolyte containing IL-TEMPO exhibits a low charging plateau at 3.75 V (Fig. 3f). There is only a trace amount of $CO_2$ detected, which is negligible, compared to that observed using the DEGDME electrolyte. The ratio of $e^-/O_2$ with IL-TEMPO during the charge process is calculated to be 1.96, which excludes the contribution of IL-TEMPO self-redox reactions. These results show that side-reactions are negligible during the discharge and charge process when IL-TEMPO is added to the electrolyte. Therefore, the discharge–charge efficiency has been dramatically improved, which in turn leads to prolonged cycle life.

**The protection of lithium metal anode**. It is important that the lithium metal anode can be protected by IL-TEMPO during cycling to avoid additional side-reactions. The electrochemical impedance spectra (EIS) of the lithium symmetric cells in Supplementary Fig. 25 indicate the formation of a stable SEI layer through the synergistic effect of TEMPO and imidazolium groups (Supplementary Note 7). Furthermore, the resting experiment of lithium–oxygen batteries in an oxygen atmosphere further confirms that the SEI layer can inhibit the corrosion of the lithium metal anode in oxygen atmosphere (Supplementary Fig. 26). The interfacial stability of lithium metal in the IL-TEMPO electrolyte was investigated by assembling lithium symmetric cells. The TEMPO electrolyte was prepared by dissolving TEMPO (10 mM) in the DEGDME electrolyte for comparison. The results (Fig. 4a) show that the over-potential in DEGDME gradually increased after 400 h, which is caused by the dendrite growth and accumulation of an unstable SEI covered unusable lithium (dead lithium). The use of bare TEMPO redox mediator deteriorates the electrochemical performance by reducing the cycle life to 310 h. In contrast, the low

discharge–charge voltages of the symmetric cell with IL-TEMPO electrolyte remain after 500 h, indicating the formation of a stable SEI layer to enable smooth lithium stripping and plating. SEM image in Fig. 4b show that the surface of lithium metal in TEMPO electrolyte has been severely corroded by the redox mediator along with large amount of lithium dendrite formation. Similar lithium dendrite growth is also found in DEGDME electrolyte after 50 cycles (Fig. 4c). On the contrary, the lithium metal manifests a smooth surface covered with a thin layer of SEI in the IL-TEMPO electrolyte (Fig. 4d). A.C. impedance tests were conducted using the lithium symmetric cells at different cycle stages. Normally, the resistance of a cell increases with cycling due to the continuous growth of SEI layers and dead lithium from the decomposition of electrolyte components (DEGDME electrolyte in Supplementary Fig. 27a). As shown in Supplementary Fig. 27b, the impedance of the cell containing IL-TEMPO increases to a certain level in the first 10 h rest, originated from the formation of an SEI layer, then quickly drops back to a lower value and maintains stabilized in the following cycles (Supplementary Note 8). This phenomenon is related to the binding of IL-TEMPO molecules to the SEI layer during cycling (Fig. 4e). The initial SEI layer is formed owing to the decomposition of the lithium salt LiTFSI and glyme molecules, while $Li^+$ and the cation of IL-TEMPO are attracted to the SEI layer functioning as the counter cations. The binding of ionic liquid molecules to the SEI layers has been previously reported during battery cycling to function as the counter cations, which can stabilize the chemical formation of an SEI layer[53–58]. Additionally, the TEMPO functional groups in the IL-TEMPO form a brush-like molecular architecture on the lithium surface (Supplementary Fig. 28). The unique n-/p-doping property of the TEMPO group can improve the interfacial ionic conductivity for the transport of $Li^+$ and electrons at different oxidation states, resulting in low resistance of the cell[59]. Moreover, the outstretched TEMPO functional group can act as a barrier to further resist the oxidation by

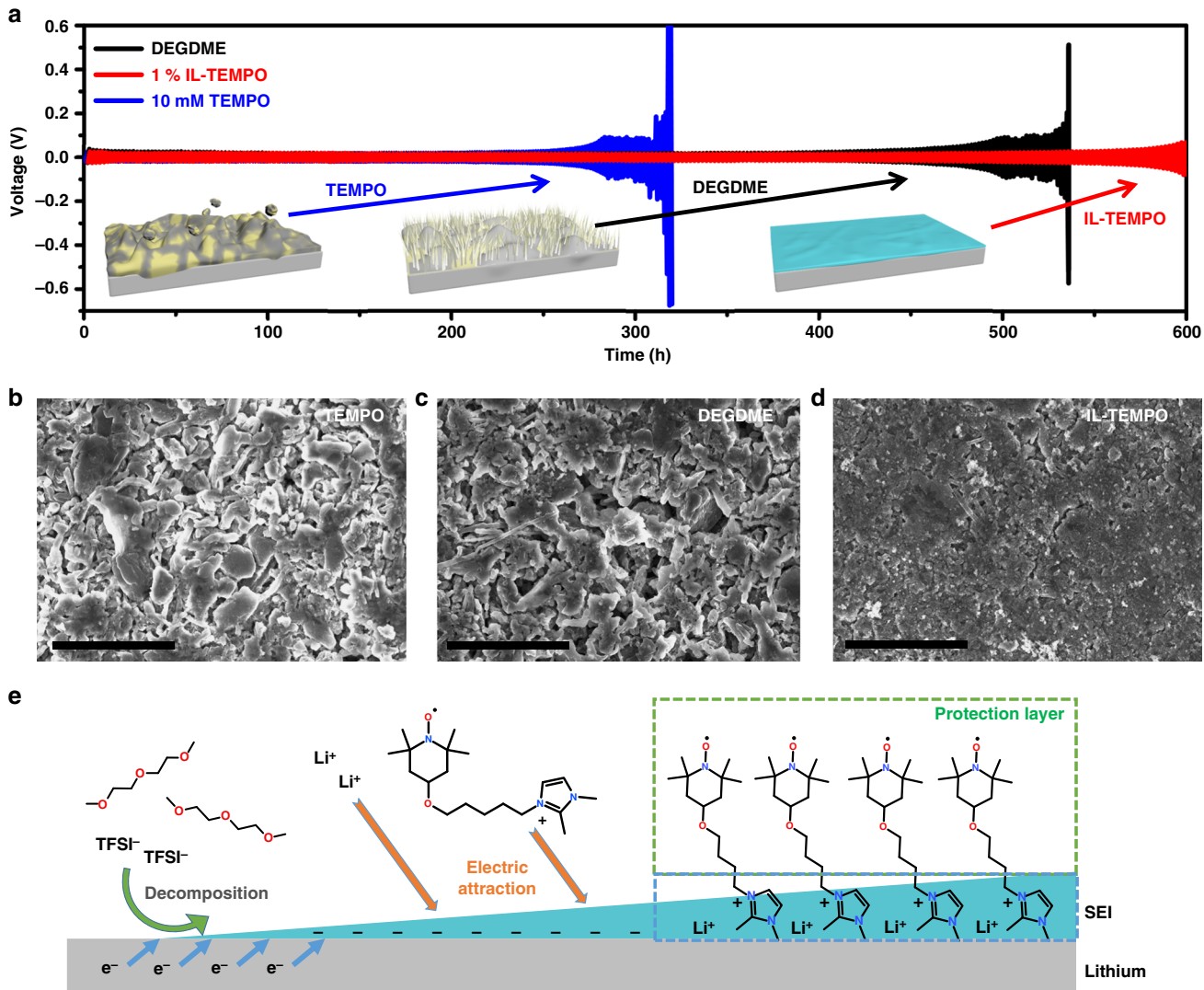

**Fig. 4** Investigation of the stability of lithium metal anode in Li|Li symmetric cells. **a** The cycling performances and voltage profiles of lithium plating/stripping in the Li|Li symmetric cells. The current density was $1\,\mathrm{mA\,cm^{-2}}$. The insets are the illustration images of the lithium anodes. **b–d** scanning electron microscope (SEM) images of the lithium metal after 50 cycles in **b** TEMPO electrolyte, **c** DEGDME electrolyte, and **d** IL-TEMPO electrolyte. Scale bars are 10 μm in **b–d**. **e** The schematic illustration of the SEI formation on lithium metal anode within the IL-TEMPO electrolyte

the dissolved oxygen and IL-TEMPO molecules from the electrolyte, resulting in higher stability of the interface. Therefore, the use of IL-TEMPO stabilizes the SEI layer to protect the lithium metal and also enhances the interfacial ionic transference, which allows much smoother lithium plating and stripping during the battery operation.

**Overall mechanism of IL-TEMPO for promoting the performance.** The functioning mechanism of IL-TEMPO is the combination of redox mediator, oxygen shuttle, and lithium protector (Fig. 1a). Firstly, the synergistic effect of the reduced N-O group and large imidazolium cation enables it to function as an oxygen shuttle, whereby the interactions with oxygen form an unstable intermediate (Supplementary Fig. 9 and Supplementary Movie 1). The intermediate carries the oxygen species to the surface of the cathode, leading to the formation of $Li_2O_2$. This process is particularly important as the combination of oxygen species with IL-TEMPO would suppress the formation of "free" $O_2^{\bullet-}$ radicals, which usually causes side-reactions

during discharge. Secondly, the highly reversible redox reaction of the TEMPO group in IL-TEMPO facilitates the oxidative decomposition of $Li_2O_2$, lowering the charge over-potential. It also increases the charge efficiency while reducing the probability of side-reactions originating from the otherwise high charging voltage. Moreover, with the assistance of the imidazolium moiety of IL-TEMPO, a more stable SEI layer with TEMPO functional groups can be formed on the surface of the lithium metal anode to prevent the corrosion of lithium metal, resulting in more facile reversible lithium stripping and plating during the operation of the Li–O₂ battery. In conclusion, IL-TEMPO possesses unique properties that allow it to comprehensively boost the electrochemical performances of Li–O₂ batteries from every aspect, hence resulting in an exceptional cycle life.

**IL-TEMPO as solvent.** The liquid IL-TEMPO is highly miscible with the DEGDME solvent, and a higher concentration of IL-TEMPO is permitted due to its unique property to protect the

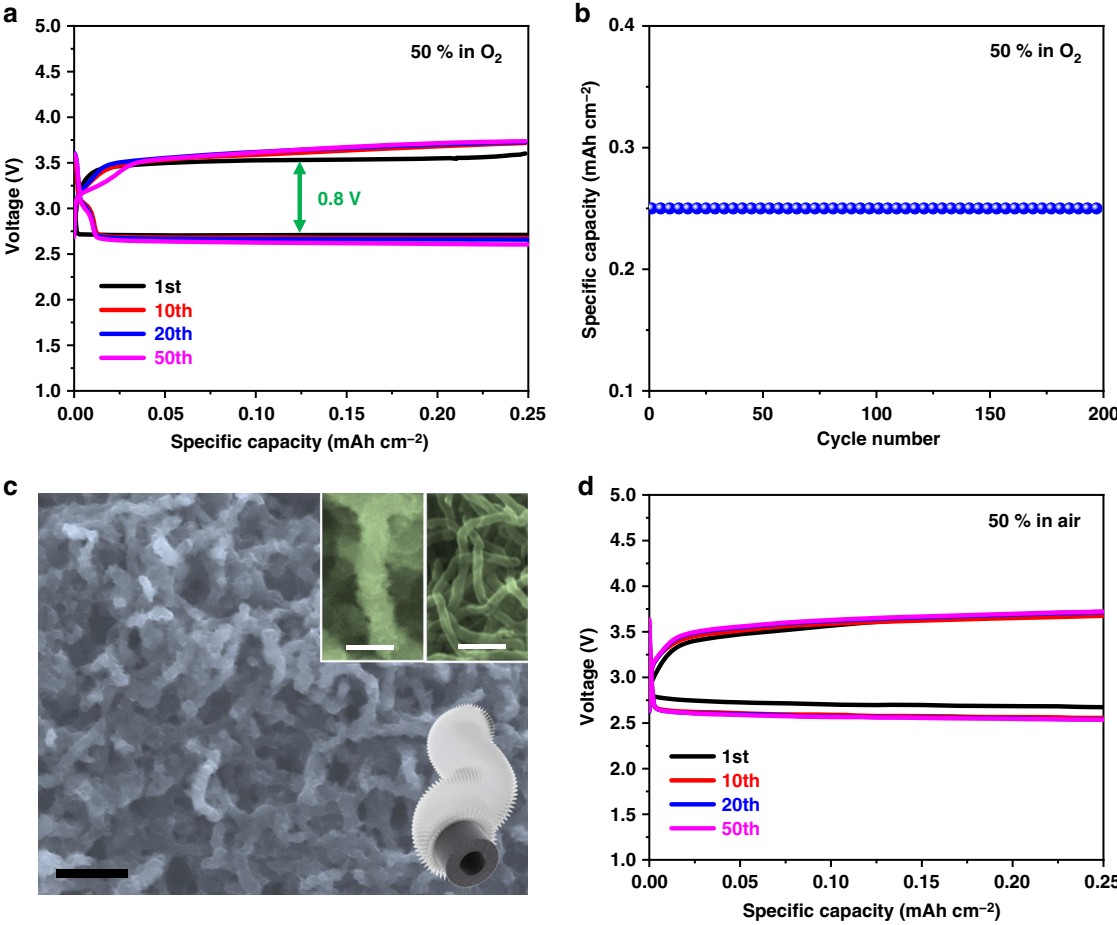

**Fig. 5** The electrochemical characterizations of IL-TEMPO (50%) when employed as electrolyte solvent. **a** The discharge–charge profiles and **b** the cycling performances of a Li–O₂ battery with the 50% electrolyte. **c** The scanning electron microscope (SEM) image of the discharged electrode wn the mixed electrolyte was used. Scale bar is 400 nm. The inset figures are the SEM image at high magnification and schematic illustration of the discharge carbon nanotube (CNT) configuration comparing to the original CNT. Scale bars are 200 nm. **d** The discharge–charge profiles of a Li–air battery with the 50% electrolyte. The electrolyte was prepared by mixing IL-TEMPO and DEGDME with a 1:1 volumetric ratio, and the concentration of LiTFSI was kept at 0.5 M. The current density was 0.1 mA cm⁻²

lithium metal anode. We found that a continuous increase of the IL-TEMPO ratio in the electrolyte to 50% could further enhance battery performance compared to lower ratios (Supplementary Fig. 29). Figure 5a shows the discharge–charge profiles of the Li–O₂ cell with the IL-TEMPO electrolyte at a 50% ratio. The discharge–charge behavior is similar to the cell with lower IL-TEMPO content (Fig. 3b) but the cycle life was significantly prolonged to more than 200 cycles (Fig. 5b). The extended cycle life is consistent with the decrease of side-reactions during discharge and charge processes. Interestingly, the XRD pattern of a discharged electrode (Supplementary Fig. 30) shows no evidence of crystalline Li₂O₂, while SEM images (Fig. 5c) show that the electrode surface is covered by small particles, which can be assigned to amorphous Li₂O₂[60]. It is further confirmed by the FTIR spectra (Supplementary Fig. 31) and titration results (Supplementary Table 2) of the electrode after discharge. As discussed above, oxygen species interact with IL-TEMPO to form unstable intermediates. The formation of the intermediate should be kinetically and thermodynamically favorable in such a high concentration of active IL-TEMPO in the electrolyte system. Therefore, high concentration intermediates could be formed during the first stage of discharge. Consequently, the formation of Li₂O₂ on the cathode would not be dominated by the electrochemical reduction of the intermediate (when the

IL-TEMPO concentration is low). We propose that amorphous Li₂O₂ is formed by the disproportionation reaction in the electrolyte. High concentration IL-TEMPO in the electrolyte should lower its solvating capability towards Li⁺, which is more favorable for the formation of amorphous Li₂O₂. The amorphous Li₂O₂ allows for easier decomposition during the charging process, thus leading to further decrease of charging over-potentials. Small quantities of intermediates should remain in the electrolyte after discharge, which would readily decompose, explaining the high discharge–charge efficiency (DEMS result in Supplementary Fig. 32 and Note 9, and titration result in Supplementary Table 2). DEGDME-free IL-TEMPO electrolyte (with 0.5 M LiTFSI) could also be used in Li–O₂ cells with the advantage that it can be operated at 70 °C due to the negligible vapor pressure of IL-TEMPO (Supplementary Figs. 33 and 34).

**IL-TEMPO electrolytes for Li–air batteries.** We further assembled the Li–air cells using the same configuration as the Li–O₂ cells, directly using air instead of pure oxygen. The electrochemical performance of the Li–air cell with the IL-TEMPO electrolyte (1%) shown in Supplementary Fig. 35a is similar to the Li–O₂ cell (Fig. 3b) with charge potentials lower than 4 V. Post-mortem XRD

characterization in Supplementary Fig. 35b illustrates that the discharge products are dominated by $Li_2O_2$, which can be reversibly decomposed during the charge process. This has been further confirmed by FTIR (Supplementary Fig. 36)[61]. Further increasing the proportion of IL-TEMPO to 50% (Fig. 5d) results in steadier discharge–charge profiles and better cycling performances. We attribute the unique properties of IL-TEMPO responsible to the successful operation of the Li–air battery. The exceptional capability of oxygen dissolution overcomes the drawback of relatively low oxygen content (21%) in air that may hinder the discharge process (Supplementary Figs. 37 and 38). Furthermore, the interferences of contaminants such as $CO_2$ and $H_2O$ are significantly reduced by the capability of IL-TEMPO to form a protective layer on the lithium anode and to form an intermediate with oxygen species. Additionally, the hydrophobic nature of the IL-TEMPO also keeps $H_2O$ away from the electrolyte, thus enhancing the overall cycle life (Fig. 5d). The development of versatile IL-TEMPO for Li–air batteries makes it possible for the realization of Li–air batteries.

## Discussion

A TEMPO-grafted ionic liquid has been synthesized and applied as a multi-functional agent in Li–$O_2$ batteries. The IL-TEMPO showed highly reversible redox reactions at 3.0 V and 3.75 V, enabling it functioning as an oxygen shuttle and redox mediator to facilitate discharge and charge processes, respectively. The imidazolium group of IL-TEMPO can further facilitate the formation of a stable SEI and smooth plating and striping of lithium on the surface of the lithium anode. Furthermore, the liquid form IL-TEMPO can function as an electrolyte solvent that showed suppressed side-reactions and further enhanced cycling performances. The combination of the unique properties allows batteries with IL-TEMPO to be operated in a harsh environment such as air atmosphere, which makes it potentially suitable for future practical applications.

## Methods

**Synthesis of IL-TEMPO**. The detailed synthesis process and characterizations of 1,2-dimethyl-3-(4-(2,2,6,6-tetramethyl-1-oxyl-4-piperidoxyl)-pentyl)imidazolium bis(trifluoromethane)sulfonimide (IL-TEMPO) is illustrated in Supplementary Fig. 1a and Supplementary Method.

**Characterizations**. The $^1$HNMR spectra were recorded on an Agilent 500 Spectrometer at 25 °C in d-DMSO. The infrared spectroscopy measurements were conducted on a Nicolet Magna 6700 FTIR spectrometer. All spectra were obtained using 4 cm$^{-1}$ resolution and 64 scans at room temperature. The spectra of ATR-FTIR were obtained using 4 cm$^{-1}$ resolution and 16 scans at room temperature with Argon protection. A field emission scanning electron microscope (FESEM, Zeiss Supra 55 VP) was employed to observe the electrode morphologies. Nitrogen-sorption measurements were carried out at 77 K with a Micromeritics 3Flex surface characterization analyzer. The specific surface area was calculated by the Brunauer-Emmett-Teller (BET) method. X-ray diffraction (XRD) measurement was performed on a Bruker D8 X-ray diffractometer using Cu Kα radiation. For the XRD analysis, the electrodes from the disassembled cells were washed and dried first, and then sealed with "Parafilm"™ to exclude moisture and carbon dioxide from the discharge products, which are very sensitive to normal atmospheric air components.

**Electrochemical characterizations**. All the electrochemical characterizations were conducted on a CH Instrument 660D electrochemical workstation and Biologic VMP3 potentiostat. The impedance spectra were measured in a frequency range of 0.01 to $10^6$ Hz. The cyclic voltammetry for two-electrode configuration was operated by using a lithium metal foil as anode, a glass fiber as separator soaking with electrolyte, and a porous electrode as cathode. Carbon paper electrodes were prepared by stacking three layers of pre-cut carbon papers (10 mm diameter, H2315, Quintech). CNT cathodes were prepared by casting the slurry of CNT and PTFE (8:2) in iso-propylene/water on the pre-cut glass fiber and dried before use. The electrolyte was prepared by dissolving IL-TEMPO in DEGDME with 0.5 M LiTFSI, and the volume concentration was 1% (IL-TEMPO/DEGDME). The scan rate was 0.5 mV s$^{-1}$.

The discharge/charge performances were evaluated by a Neware Battery Testing System. The discharge–charge performances were evaluated by assembling Li–$O_2$

batteries. A two-electrode Swagelok-type cell with an air hole (0.785 cm$^2$) on the cathode side was used to test the electrochemical performances. The cells were assembled in an argon filled glove box with water and oxygen level <0.1 ppm. Lithium foil was used as the anode and the glass fiber was used as the separator. The electrolytes were prepared by dissolving 0.5 M LiTFSI in DEGDME with different volume ratio of IL-TEMPO to DEGDME. Pure IL-TEMPO electrolyte was prepared by dissolving 0.5 M LiTFSI in IL-TEMPO. The typical amount of electrolyte used in a single cell was 150 μL. The assembled cell was gas tight except for the cathode side window, which is exposed to the oxygen tank. All measurements of Li–$O_2$ batteries were conducted in a gas tank of 1 atm dry oxygen atmosphere. And all the Li–air batteries were measured through the same configuration with Li–$O_2$ batteries with a tank of regular air by purging air in the tank for 30 min The capacities of the batteries were calculated based on the total area of the electrodes.

The Li|Li symmetrical cells were assembled using 2032 type coin cell. The lithium metals were pre-cut to 10 mm diameter. Glass fiber separators were used to absorb electrolytes. The current density was 0.1 mA cm$^{-2}$.

A modified Swagelok-type Li–$O_2$ cell was linked to a commercial magnetic sector mass spectrometer (Thermo Fischer) by a specially designed gas-purging system for in situ DEMS measurement. The flow rate of purge gas was set at 1 mL min$^{-1}$. During the discharge process, a mixture of Ar/$O_2$ (mass ratio 1:4) was used as the carrier gas to observe the oxygen consumption. Ar acts as the internal trace amount of gas with a known constant flux. During the charge process, high-purity Ar was used as the carrier gas. The DEMS cells were assembled in the Ar-filled glove box for the electrochemical testing.

**Demonstration experiments**. The demonstration experiments of the catalytic capability of IL-TEMPO towards $Li_2O_2$ and $Li_2CO_3$ were conducted by directly charging the cells with electrodes loaded with $Li_2O_2$ or $Li_2CO_3$. The detailed preparation process is in the Supplementary Method.

The demonstration experiments of interactions between IL-TEMPO and oxygen were conducted by altering the electrochemical reduction of IL-TEMPO into chemical reduction with the aid of phenylhydrazine (PhNHNH$_2$) and N,N-dimethylacetamide (DMA). The detailed experimental process is in the Supplementary Method.

**$Li_2O_2$ titration method**. The yields of $Li_2O_2$ after the discharge process of Li–$O_2$ batteries are measured through a $Li_2O_2$ titration method, which is reported previously in the literature[52]. Detailed experimental process is illustrated in the Supplementary Method.

## Data availability

The data that support the findings of this study are available from the corresponding authors upon reasonable request.

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

## Acknowledgements

This project is financially supported by the Australian Research Council (ARC) through the ARC Discovery Project (DP160104340) and the Rail Manufacturing Cooperative Research Center (RMCRC) project. We also acknowledge the support from the Start-up Grant from Dongguan University of Technology (DGUT) for High-Level Talent Program (KCYKYQD2017016), the China Postdoctoral Science Foundation (Project No. 2017M622682), and Research Start-up Funds of DGUT (GC300501-10).

## Author contributions

J.Z. and G.W. conceived the idea. J.Z. performed the electrochemical experiments. J.Z. and A.T. synthesized the IL-TEMPO. J.Z. and K.Y. performed experiments on lithium anode stability tests. J.Z., Y.Z., and X.G. conducted characterizations. Z.L. performed DEMS experiments and analysis. B.S., A.M.M., D.S., T.R., C.W., M.A., and Z.P. were all involved in discussions. All of the authors contributed to the writing to the manuscript before submission.

## Additional information

**Competing interests:** The authors declare no competing interests.

