## [Peer Review File · Nature Communications]

Reviewers' comments:

Reviewer #1 (Remarks to the Author):

Authors reported a TEMPO-based ionic liquid (IL-TEMPO) as multi-functional electrolyte for Li-O₂ batteries. The authors claimed that this ionic liquid promoted solution discharge, mediated oxidation process and protected lithium anode, which led to the much-improved cell performance. The results are quite encouraging and the idea is new, however, the evidence is not solid enough. Some questions in experiment design and data analysis need to be clarified before publication.

1. Authors synthesised IL-TEMPO and confirmed its structure in line 90-98. However, explanation and clarification of FTIR peaks is missing in the supporting information. Can authors comment on purity of IL-TEMPO, as it is well-acknowledged ionic liquid is difficult to purify and impurities might affect the cell performance.
2. Authors claimed in CV in Fig.1b, peaks at 3.0V corresponded to the reduction of N-O radicals, however, the first reduction and oxidation peak in CV in Fig. 1b is lower than 3.0 V and the reversibility is poor. Authors need to clarify it in the paper. If the difference from the second scan is due to the formation of SEI, authors can use a lithium anode with the SEI layer formed and redo CV measurements to prove their assumption.
3. Authors used XRD to characterise discharged and charged cathodes while SEM was applied only on discharged cathode. Please add SEM images of charged electrodes to prove the oxidation. As is well known, side products like organic carbonates might be amorphous and cannot be detected by XRD. Please add FTIR data after discharge and charge to compensate in O₂ and air systems. It would also be beneficial if authors can add characterisations after 10th and 50th cycles.
4. More detailed explanation of impedance data will be more convincing. The authors also said 'Figure 4 b-d show a large amount of lithium dendrites in DEGDME', however, it is difficult to recognise in the figures. Can authors emphasize it in the figure. The lithium dendrites in IL-TEMPO slight contradicts with the theory of SEI formation. Can authors explain it in the paper?
5. Authors did not detect any crystallised Li₂O₂ in 50% ratio IL-TEMPO and then ascribed it to amorphous Li₂O₂, which is doubtful. It is commonly accepted that solution discharge leads to large particles formation rather than amorphous morphology. Authors should at least include some evidence to prove formation of Li₂O₂, such as FTIR, XPS.
6. Stability is a big concern in this field, especially TEMPO species as a free radical. After 50 or 200 cycles, what is the percentage of IL-TEMPO leftover and can authors comment on decomposition of electrolytes.
7. It is important to quantify the amount of Li₂O₂ during discharge. Can authors add the yield measurements (DOI: 10.1021/jz401659f) to quantify the amount under O₂, 50% IL-TEMPO and air systems?

Reviewer #2 (Remarks to the Author):

I have carefully reviewed the paper by Zhang et al., concerning the development of IL-TEMPO as a potential redox mediator and also as a designer electrolyte solvent for Li-O₂ and Li-air batteries. The area of research is exciting and the designer electrolytes for Li-O₂ have been called for by several researchers in the field as conventional solvents seem to not be stable within the electrochemical operating conditions of Li-O₂ and Li-air batteries. In fact, this seems to be true for other alkali metal-oxygen chemistries as well. Therefore, I think that this paper warrants publication in Nature Communications. However, I have the following serious concerns and therefore advise that the paper be considered for publication after the following are fully addressed:

1) The authors in Page 6 line 113-115 suggest that the shift in the N-O reduction potential is possibly due to SEI formation on the Lithium Anode during electrochemical CV measurements. I suggest that Impedance data on the Li Anode to corroborate this hypothesis be provided.

2) A major concern in the paper is the redox potentials for IL-Tempo and Li-O₂ cells. The two prominent redox peaks in the CV for IL-TEMPO are at 3 V and 3.6 V. A nearly flat potential during a galvanostatic charge step of Li-O₂ cells being around 3.5-3.6 V is reasonable assuming redox mediation by IL-TEMPO. However, the galvanostatic discharge potential for Li-O₂ cells is ~ 2.7 V and redox mediation by IL-TEMPO during discharge is not expected. The authors should clearly explain this apparent discrepancy.

3) The authors have performed CVs in an inert Argon atmosphere to show redox behavior of IL-TEMPO. However, a more pertinent and useful measurement is to perform CVs using inert electrodes in the presence of oxygen. This is necessary to understand and irreversibility inherent to IL-TEMPO in the presence of oxygen which is the more realistic operating condition. Unless this experiment, the authors cannot conclude beyond doubt that the contribution of IL-TEMPO self-redox to capacity is negligible.

4) The very small discharge capacity of Li-O₂ cells in a DEGDM electrolyte without IL-TEMPO is not expected. The authors must compare their ultimate areal and gravimetric capacities with other published work to show that the capacities obtained in their measurements are not much lower than published work.

5) X-ray diffractograms presented in this manuscript are in general not rigorously indexed. I suggest that all XRD patterns be indexed according to standard JCPDS patterns and ensure that the peak positions and intensity ratios are in agreement with JCPDS standards. This is very much necessary as some of the high intensity peaks for LiOH and Li₂O₂ are very close in 2-Theta values.

6) According to Figure 3, the DEMS data shows that the Li-O₂ cells with IL-TEMPO evolve more oxygen than they consume during discharge. Isn't this a serious issue? What is the error bar on these measurements?

7) The authors have used a capacity of 0.25 mAh/cm² for cycling measurements. This is more than 15 times smaller than the ultimate capacity of the cells. A common standard in battery research is to cycle these cells to at least 80% of their full capacity. While cyclability at a depth of discharge of at least 80% is not necessary publishing this work, such measurements reveal the limitations of the chemistry and provide inspiration for future work.

8) The authors seem to have incorrect units in certain figures and captions. For example Fig. S25 and caption in Fig. 3. gravimetric normalization is used in these two instances while the rest of the paper uses areal normalization.

9) Page 15 Lines 312 - 321: The authors suggest that the absence of Li₂O₂ peaks in XRD is indicative of amorphous Li₂O₂. However, there is no evidence for this assumption. For example, the authors could perform peroxide titrations or Raman/FTIR measurements to confirm the presence of Li₂O₂. Without such evidence, it will be hard to convince the readers of the presence of Li₂O₂

10) Page 16 Line 346 - 348: There is no evidence anywhere in the paper to suggest that these batteries could work at elevated temperatures of upto 70 C. I suggest that the authors delete this sentence from the concluding paragraph.

11) Figs 5d, S24 and S27 are contradicting Figures S15. While the authors show that Li_2CO_3 is not oxidized by IL-TEMPO, the experiments in Li-Air cells where carbonate formation is expected suggest that there is negligible difference between the galvanostatic charge curves for Li-O₂ and Li-air cells. This is evidence that Li_2CO_3 is likely oxidizing at around 3.6 V. Also, Fig. S24 clearly shows a small but observable increase in CO₂ evolution. The authors need to address this discrepancy.

12) Finally, none of the galvanostatic charge curves polarize similar to the polarization observed in Fig. S15. A potential polarization is expected at the end of charge for any electrochemical devices. The authors must address such as polarization is not observed in all of their limited capacity cycling data.

Reviewer #1 (Remarks to the Author):

Authors reported a TEMPO-based ionic liquid (IL-TEMPO) as multi-functional electrolyte for Li-O₂ batteries. The authors claimed that this ionic liquid promoted solution discharge, mediated oxidation process and protected lithium anode, which led to the much-improved cell performance. The results are quite encouraging and the idea is new, however, the evidence is not solid enough. Some questions in experiment design and data analysis need to be clarified before publication.

Comment 1: *Authors synthesised IL-TEMPO and confirmed its structure in line 90-98. However, explanation and clarification of FTIR peaks is missing in the supporting information. Can authors comment on purity of IL-TEMPO, as it is well-acknowledged ionic liquid is difficult to purify and impurities might affect the cell performance.*

Response 1: We would like to thank the reviewer for the valuable suggestion. Following the reviewer's comment, we have added the explanation and clarification of FTIR peaks of IL-TEMPO in the revised Supplementary Information.

Figure R1 | FTIR spectrum of the IL-TEMPO.

The FTIR of IL-TEMPO shows the characteristic peaks of TFSI anion and imidazolium ionic liquids. FTIR ATR (n, cm^{-1}) is analysed as: 571 (CF₃ asym bend), 610, 615 (SO₂ asym bend),

650 (S-N-S bend), 741 (CF₃ sym bend), 762, 789 (skeletal asym bend of imidazolium ring), 1056 (S-N-S asym str, skeletal asym str of imidazolium ring, C-C asym str, N-CH₃ twisting, aliphatic ether C-O str), 1136 (SO₂ sym str), 1180 (N-CH₂ and N-CH₃ C-N str, C-C str, CF₃ asym str), 1226 (CF₃ sym str), 1331, 1348 (SO₂ asym str), 1462, 1539, 1590 (skeletal str of imidazolium ring), 2869, 2939, 2976 (sp³ C-H str), 3148 (sp² C-H str).

The purification of IL-TEMPO was conducted by using column chromatography and confirmed by using thin-layer chromatography (TLC) and NMR technique. During the synthesis process, when 1,5-dibromopentane reacted with 4-hydroxy-TEMPO, the resulting 5-TEMPO-pentyl bromide was purified by column chromatography on silica. At this stage, the purity of the compound was examined by TLC and ¹H NMR data. All NMR spectra were recorded with the addition of phenylhydrazine, to reduce a paramagnetic N-O group to a diamagnetic N-OH.

After quaterisation reaction with 1,2-dimethylimidazole, the impurities were removed by trituration with diethyl ether followed by filtration. The purity of the compound was again confirmed by TLC and ¹H NMR:

¹H NMR (DMSO-*d*₆, ppm): *d* = 1.04 (s, 6H, CH₃), 1.07 (s, 6H, CH₃), 1.19–1.34 (m, 4H, C(3)_{Pip}-H and C(5)_{Pip}-H, -CH₂-CH₂-CH₂-N<), 1.46–1.52 (m, 2H, -CH₂-CH₂-N<), 1.68–1.76 (m, 2H, -O-CH₂-CH₂-), 1.79–1.86 (m, 2H, C(3)_{Pip}-H and C(5)_{Pip}-H), 2.58 (s, 3H, C(2)_{Im}-CH₃), 3.36 (t, 2H, ³*J*(H,H) = 6.5 Hz, -O-CH₂-), 3.45–3.54 (m, 1H, C(4)_{Pip}-H), 3.75 (s, 3H, N(1)_{Im}-CH₃), 4.10 (t, 2H, ³*J*(H,H) = 7.3 Hz, -CH₂-N≤), 7.62–7.65 (m, 1H, C(4)_{Im}-H or C(5)_{Im}-H), 7.65–7.67 (m, 1H, C(4)_{Im}-H or C(5)_{Im}-H).

After the exchange of Br⁻ to TFSI⁻, the obtained IL-TEMPO was fully characterised. FTIR of IL-TEMPO confirms the unique ionic liquid structure (Figure R1). More importantly, IL-TEMPO was characterised by ¹³C, ¹H NMR (Figure S1), and elemental analysis, which proved the absence of by-products in the bulk phase of this compound. NMR spectra of IL-TEMPO indicated a highly pure compound. The additional peaks observed on NMR are related to the oxidised phenylhydrazine, products of its decomposition and water in the DMSO-*d*₆.

¹H NMR (DMSO-*d*₆, ppm): *d* = 1.04 (s, 6H, (CH₃)_{Pyp}), 1.08 (s, 6H, (CH₃)_{Pyp}), 1.19–1.34 (m, 4H), 1.46–1.52 (m, 2H), 1.67–1.75 (m, 2H), 1.80–1.87 (m, 2H), 2.54–2.59 (m, 3H, C(2)_{Im}-

CH₃), 3.37 (t, 2H, ³J(H,H) = 6.4 Hz, -O-CH₂-), 3.45–3.55 (m, 1H, C(4)_{Pyp}-H), 3.71–3.76 (m, 3H, N(1)_{Im}-CH₃), 4.09 (t, 2H, ³J(H,H) = 7.6 Hz, -CH₂-N≤), 7.58–7.61 (m, 1H, C(4)_{Im}-H or C(5)_{Im}-H), 7.61–7.64 (m, 1H, C(4)_{Im}-H or C(5)_{Im}-H).

¹³C NMR (DMSO-*d*₆, ppm): *d* = 9.18 (C(2)_{Im}-CH₃), 20.83 ((CH₃)_{Pip}), 22.76 (-CH₂-(CH₂)₂-N<), 29.19 (-CH₂-CH₂-N<), 29.34 (-O-CH₂-CH₂-), 32.60 ((CH₃)_{Pip}), 34.75 (N(1)_{Im}-CH₃), 44.93 (C(3)_{Pip}, C(5)_{Pip}), 47.70(-CH₂-N≤), 58.19 (C(2)_{Pip}, C(6)_{Pip}), 67.19 (-O-CH₂-), 70.31 (C(4)_{Pip}), 119.82 (q, CF₃, ¹J(C,F) = 322 Hz), 120.96, 122.43 (C(4)_{Im}, C(5)_{Im}), 144.12 (C(2)_{Im}).

The result of elemental analysis is shown below. During elemental analysis, the compounds were not stored in vacuum, which is why they absorbed some water. It was removed before the battery assembly by drying the compounds in vacuum at 100 °C for 24 hrs before transferring to the glovebox.

Anal. Calcd for C₂₁H₃₅F₆N₄O₆S₂·1.5H₂O: C 39.12, H 5.94, N 8.69. Found: C 39.42, H 5.74, N 8.26.

Therefore, it can be concluded that the synthesized IL-TEMPO is pure for the employment in the Li-O₂ batteries.

The ¹H NMR of Br-ionic liquid and elemental analysis of IL-TEMPO have been added in the experimental section on Page 2-3 in the revised Supplementary Information. The revised FTIR spectrum and the peak index have been added as Figure S2 on Page 7 in the revised Supplementary Information. The clarification of structure and purity of IL-TEMPO have been added on Page 4 in the revised Manuscript.

IL-TEMPO was characterised by hydrogen nuclear magnetic resonance (¹H NMR, Figure S1b) and Fourier transform infrared spectroscopy (FTIR, Figure S2), to confirm its molecular structure and high purity.

Comment 2: *Authors claimed in CV in Fig. 1b, peaks at 3.0V corresponded to the reduction of N-O radicals, however, the first reduction and oxidation peak in CV in Fig. 1b is lower than 3.0 V and the reversibility is poor. Authors need to clarify it in the paper. If the difference from the second scan is due to the formation of SEI, authors can use a lithium anode with the SEI*

layer formed and redo CV measurements to prove their assumption.

Response 2: We would like to thank the reviewer for the valuable suggestion. As shown in Figure S3a (original Figure 1b), the first reduction and oxidation peaks beneath 3 V correspond to the reversible reduction of IL-TEMPO, and the second reduction and oxidation peaks above 3 V are the reversible oxidation of IL-TEMPO. Comparing to the first cycle, the pair of peaks corresponding to the oxidation of IL-TEMPO does not change for 3 cycles, proving its reversibility. However, the peaks related to the reduction of IL-TEMPO both shift positively in the following cycles. The possible reason is that during the first cycle, IL-TEMPO participates in the SEI formation on the lithium metal, forming an IL-TEMPO protection layer. The reaction leads to slightly change of the lithium reference potential and conductivity, causing the shift of the reduction potential. The SEI formation mostly completed after the reduction process, which was proven by the unchanged potential of oxidation potential.

Following the reviewer's suggestion, a demonstration experiment has been conducted by using SEI-formed lithium anode to bypass the SEI formation process. The SEI layer is formed by soaking the lithium metal in the IL-TEMPO contained propylene carbonate (PC) electrolyte (LiTFSI was added) for 5 days to simulate the SEI formation during the CV test. The result is shown below as Figure R2. Compared to the original CV curves in Figure S3a (original Figure 1b), there is no significant peak shift in the CV curves obtained with SEI-formed lithium anode, demonstrating that the shift in the first cycle in the original CV curves is mainly caused by the formation of SEI layer.

Figure R2 | Cyclic voltammetry curve of battery with IL-TEMPO electrolyte in argon atmosphere. The scan rate is 0.5 mV s^{-1} . The inset image is the illustration of

reduction/oxidation of IL-TEMPO. The lithium anode is pre-treated by soaking in IL-TEMPO containing propylene carbonate solution for 5 days.

Moreover, electrochemical impedance spectroscopy (EIS) spectra were measured after each cycle of the cell with unprotected lithium anode to further confirm the mechanism (Figure R3). The a.c. impedance of the cell slightly increases after the first cycle and stabilizes in the following cycles. This phenomenon proves that SEI layers are formed during the first cycle, which would stabilize the lithium anode, leading to highly reversible redox reactions. The incorporation of IL-TEMPO molecules in the SEI layer leads to a relatively low interface resistance because the existence of IL-TEMPO could efficiently enhance the ionic transport through the SEI layer. Therefore, the possible reason of the shift of the first pair of reduction peaks is the formation of IL-TEMPO-incorporated SEI layer during the first cycle.

Figure R3 | The impedance spectra of the cell after each cycle during the CV test. The lithium metal is not pre-treated to form SEI layer in advance.

The newly measured CV curves have been added as Figure 1b on Page 5 in the revised Manuscript. The a.c impedance spectra and the original CV curves have been added as Figure S3 on Page 8 in the revised Supplementary Information. The description has been added on Page 8 in the revised Supplementary Information.

The potential for the reduction of the N-O radical undergoes a positive shift after the first cycle

as shown in Figure S3a, which could be due to the formation of a solid electrolyte interface (SEI) on the surface of the lithium anode. This has been proven by monitoring the a.c. impedance spectra change after each cycles in Figure S3b. Meanwhile, the CV profiles in the second and third cycles (Figure S3a) match perfectly with the CV curves obtained using pre-treated lithium anode (Figure 1b), further confirming the SEI formation during the CV tests.

Comment 3: *Authors used XRD to characterise discharged and charged cathodes while SEM was applied only on discharged cathode. Please add SEM images of charged electrodes to prove the oxidation. As is well known, side products like organic carbonates might be amorphous and cannot be detected by XRD. Please add FTIR data after discharge and charge to compensate in O₂ and air systems. It would also be beneficial if authors can add characterisations after 10th and 50th cycles.*

Response 3: We would like to thank the reviewer for the valuable suggestion. Following the reviewer's suggestion, we have performed the post-mortem SEM characterization on the charged carbon paper electrode and the SEM images are shown below as Figure R4. The images clearly show that there is no residue discharge product left after charge process and the morphologies of the carbon papers have recovered back to their original states. This indicates the discharge and charge processes of the Li-O₂ batteries are highly reversible, and no visible side-products are observed after the first cycle.

Figure R4 | The SEM images of the carbon paper electrode after the first charge process from the Li-O₂ cells with a. DEGDME electrolyte and b. 1 % IL-TEMPO electrolyte.

We have also performed SEM characterization on the charged CNT electrode (Figure R5) which also proves the reversible formation/decomposition of Li₂O₂ on the CNT electrodes.

Figure R5 | The SEM images of the CNT electrode after the first charge process from the Li-O₂ cells with 1 % IL-TEMPO electrolyte.

Side-reactions often occur during the operation of Li-O₂ batteries. During our experiments, XRD has been employed to prove that the discharge products are dominated by Li₂O₂ and the reaction is highly reversible. Following the reviewer’s suggestion, FTIR measurement has been performed on the discharge and charge electrodes to further identify the reversibility of the discharge products. The as-obtained FTIR spectra are shown below in Figure R6. Similar to the XRD results, the discharge products are dominated by Li₂O₂, and no obvious peaks can be assigned to the formation of side-products such as organic carbonates and Li₂CO₃ can be observed. The peaks assigned to Li₂O₂ disappear after the charging process, indicating the highly reversible reaction. Additionally, the FTIR spectra do not significantly change after 10 cycles, which also proves that there are very limited side-reactions occurring during the battery operations.

Figure R6 | The FTIR spectra of the electrode before and after cycles using the 1 % IL-TEMPO electrolyte. **a.** The cells were operated in O₂ atmosphere. **b.** The cells were operated in air atmosphere.

Moreover, we performed the titration experiment on the discharged electrode. The results also confirm the assertion that minimum of side-products are formed while the discharge products are dominated by Li_2O_2 .

Table R1 Discharge Li_2O_2 percent yield ($Y_{\text{Li}_2\text{O}_2}$) for the cells with 1 % IL-TEMPO electrolyte under O_2 and air atmosphere

Electrolyte	Atmosphere	Capacity (mAh)	Li_2O_2 (μmol)	$Y_{\text{Li}_2\text{O}_2}$ (%)
1 % IL-TEMPO	O_2	2.08±0.02	36±1	92.7±1.7
	Air	2.08±0.02	32±2	82.6±4.5

The SEM images of the charged electrodes have been added as Figure S8 on Page 13 and Figure S17c on Page 22 in the revised Supplementary Information. The FTIR spectra have been added as Figure S18 on Page 23 and Figure S36 on Page 41 in the revised Supplementary Information. The corresponding explanations have been added in the revised Manuscript.

Page 7: The electrode could recover to its original state after charge, indicating the high reversibility (Figure S8).

Page 10: Thus, the capacity mainly originates from the reversible formation and decomposition of the Li_2O_2 discharge product (confirmed by XRD, SEM, and FTIR in Figure S17-18 and demonstration experiment in Figure S19), which is facilitated by the reversible redox activities of IL-TEMPO.

Page 15-16: Post-mortem XRD characterization in Figure S35b illustrates that the discharge products are dominated by Li_2O_2 , which can be reversibly decomposed during the charge process. This has been further confirmed by FTIR (Figure S36).

Comment 4: *More detailed explanation of impedance data will be more convincing. The authors also said ‘Figure 4 b-d show a large amount of lithium dendrites in DEGDME’, however, it is difficult to recognise in the figures. Can authors emphasize it in the figure? The lithium dendrites in IL-TEMPO slight contradicts with the theory of SEI formation. Can authors explain it in the paper?*

Response 4: We would like to thank the reviewer for the valuable comment. The impedance

data are used to provide evidence for influence of IL-TEMPO on the formation and electrochemical property of the SEI layer on the lithium anode. The impedance of Li|IL-TEMPO Electrolyte|Li symmetric cell in Figure S25b increases with the resting time and stabilizes after around 50 h. Compared to the impedance change in the Li|DEGDME Electrolyte|Li cell in Figure S25a, it proves that IL-TEMPO could facilitate the formation of SEI layer on the lithium anode. The process can be further accelerated when the Li-O₂ cells are resting in O₂ atmosphere when the impedances of both cells dramatically increase in the first 20 h due to the participation of O₂ (Figure S26). It is worth noting that the impedance of the cell with IL-TEMPO electrolyte stabilizes after a few hours resting while the impedance of DEGDME electrolyte continuously increase even after 80 h, which demonstrates that the as-formed SEI layer in the IL-TEMPO electrolyte can protect the lithium anode from further corrosion. The possible reason is the incorporation of IL-TEMPO molecules in the SEI layer to form protective brush-like protection to prevent the further contact of IL-TEMPO and O₂ molecules. As shown in Figure S27, the impedance of the cell with IL-TEMPO electrolyte increases in the first 10 h rest due to the formation of SEI layer, and decreases during cycling. This phenomenon indicates that the incorporation of IL-TEMPO molecules can be accelerated during the electrochemical process, and the existence of IL-TEMPO can significantly enhance the ion transport through the SEI layer, which can be seen by comparing the impedance spectra in Figure S27. In conclusion, the addition of IL-TEMPO in the DEGDME electrolyte can facilitate the formation of SEI layer and the IL-TEMPO molecules could be incorporated in the SEI layer to enhance the ion transport while protecting the lithium metal from further corrosion.

The use of IL-TEMPO during the operation of Li|Electrolyte|Li symmetric cell can significantly enhance the stability of the lithium anodes. The SEM images shown in Figure 4b-d provide a visualised result of the lithium metal surfaces in different electrolyte systems after the symmetric cell tests. Figure 4b-d show the SEM images of the lithium surfaces from cells with different electrolytes after 50 cycles. Figure 4b shows the lithium surface cycled from the TEMPO electrolyte, which has been covered by large amount of particles- and branches-like structures referring to the lithium dendrites and dead lithium. This is due to the incapability of DEGDME electrolyte to form stable SEI layer to prevent the formation of lithium dendrites and the corrosion of lithium metal by the continuous attack by the TEMPO free radicals. Similar structures can also be found in the DEGDME electrolyte due to the lack of stable protection layer (Figure 4c). However, in Figure 4d the surface of the lithium metal in IL-TEMPO electrolyte shows a relatively smooth morphology with minimum lithium dendrites

and a thin layer of stable SEI layer covering the surface, which is then proven by the EIS spectra (Figure S27) and electrochemical performance of the lithium symmetric cell (Figure 4a). The incorporation of IL-TEMPO molecules in the SEI layer forms a brush-like protection layer which can efficiently prevent the further contact between the lithium metal and free IL-TEMPO molecules (Figure 4e). This could explain the smooth surface and minimum degree of corrosion. These results are consistent with the theory of stable SEI formation with the aid of IL-TEMPO.

Figure R7 | Investigation of the stability of lithium metal anode in Li|Li symmetric cells.

a. The cycling performances and voltage profiles of lithium plating/stripping in the Li|Li symmetric cells. The current density was 0.1 mA cm^{-2} . The insets are the illustration images of the lithium anodes. **b-d.** SEM images of the lithium metal after 50 cycles in **b.** TEMPO electrolyte, **c.** DEGDME electrolyte and **d.** IL-TEMPO electrolyte. **e.** The schematic illustration of the SEI formation on lithium metal anode within the IL-TEMPO electrolyte.

Figure 4 has been revised in order to emphasize the SEM morphologies of the lithium anodes obtained in different electrolytes. The explanation section of the SEM images on Page 12 in

the revised Manuscript has been revised for better addressing the idea.

SEM image in Figure 4b show that the surface of lithium metal in TEMPO electrolyte has been severely corroded by the redox mediator along with large amount of lithium dendrite formation. Similar lithium dendrite growth is also found in DEGDME electrolyte after 50 cycles (Figure 4c). On the contrary, the lithium metal manifests a smooth surface covered with a thin layer of SEI in the IL-TEMPO electrolyte (Figure 4d).

Comment 5: *Authors did not detect any crystallised Li_2O_2 in 50% ratio IL-TEMPO and then ascribed it to amorphous Li_2O_2 , which is doubtful. It is commonly accepted that solution discharge leads to large particles formation rather than amorphous morphology. Authors should at least include some evidence to prove formation of Li_2O_2 , such as FTIR, XPS.*

Response 5: We would like to thank the reviewer for the valuable comment. IL-TEMPO has been demonstrated to participate in the ORR to interact with the intermediate superoxide radicals ($\text{O}_2^{\cdot-}$), functioning as a redox shuttle to facilitate the discharge process. With the increase of the concentration of IL-TEMPO, this process can be significantly enhanced, leading to the formation of large quantity of IL-TEMPO- O_2 intermediate. Meanwhile, the increase of IL-TEMPO proportion could significantly reduce the Li^+ solvation capability of the DEGDME electrolyte due to the introduction of the large imidazolium cation, which could reduce the buffered reactivity of Li^+ ion towards $\text{O}_2^{\cdot-}$. The combination could lead to the formation of Li_2O_2 with smaller size or even amorphous phase (*Angew. Chem. Int. Ed.* 55, 10717, 2016).

When characterizing the discharge product from 50 % ratio IL-TEMPO electrolyte, the XRD pattern of the discharged electrode shows no visible peaks associated with crystalline Li_2O_2 . However, the *in-situ* DEMS result in Figure S32 shows that oxygen releases during the discharge process, which should originate from the decomposition of Li_2O_2 . Following the reviewer's suggestion, we performed FTIR (Figure R8) and titration (Table R2), which further confirmed the existence of Li_2O_2 in the discharged electrode. Therefore, it can be concluded that Li_2O_2 with amorphous nature is the main discharge product.

Table R2 Discharge Li_2O_2 percent yield ($Y_{\text{Li}_2\text{O}_2}$) for the cells with 50% IL-TEMPO electrolyte under O_2 atmosphere

Electrolyte	Atmosphere	Capacity (mAh)	Li ₂ O ₂ (μmol)	Y _{Li₂O₂} (%)
50 % IL-TEMPO	O ₂	1.10±0.02	13±5	63.2±24.1

Figure R8 | FTIR spectra of the electrodes before and after cycles using the 50 % IL-TEMPO electrolyte.

The titration results have been added as Table S2 at Page 46 and the FTIR spectra have been added as Figure S31 at Page 36 in the revised Supplementary Information. The relevant description has been added at Page 15 in the revised Manuscript.

It is further confirmed by the FTIR spectra (Figure S31) and titration results (Table S2) of the electrode after discharge.

Comment 6: *Stability is a big concern in this field, especially TEMPO species as a free radical. After 50 or 200 cycles, what is the percentage of IL-TEMPO leftover and can authors comment on decomposition of electrolytes.*

Response 6: We would like to thank the reviewer for the valuable comment. Eliminating side-reactions of Li-O₂ batteries is crucial for achieving high electrochemical performance and long cycle life. The most commonly recognized causes of side-reactions during the operation of Li-O₂ batteries are the decomposition of electrolyte (caused by superoxide attack and high

charging voltage) and corrosion of lithium anode (caused by dissolved oxygen, water and redox mediator). In this study, we proposed a functional TEMPO-grafted ionic liquid as an oxygen shuttle, redox mediator, and a lithium metal protector to enhance the discharge process, the charge process, and the lithium anode. During the discharge process, IL-TEMPO could interact with the reduced superoxide, significantly reducing the risk of it attacking the electrolyte molecules. During the charge process, the over-potential can be dramatically reduced to 3.6 V, resulting a minimum of electrolyte decomposition and high round-trip efficiency. Moreover, the capability of forming stable SEI layer can efficiently prevent the direct contact between the dissolved species and the active lithium metal, while still maintaining fast ion transport. Therefore, the addition of IL-TEMPO in the electrolyte system can significantly enhance the stability of the Li-O₂ batteries, thus leading to a prolonged cycle life.

In order to determine the leftover of IL-TEMPO, we employed ultraviolet (UV) to determine the change of IL-TEMPO content. The samples were prepared by soaking the electrolyte-adsorbed glass fibre separator in 3 ml DEGDME solvent, and the solutions were directly used for UV tests. The UV spectrum of the electrolyte after the first cycle shows that the concentration of IL-TEMPO slightly reduces, due to the incorporation of small amount of IL-TEMPO during the formation of SEI on the lithium anode. However, the concentration of IL-TEMPO stabilizes during the following cycles. The concentration reaches almost the same as the one after first cycle. There is no additional peaks observed in the UV spectra. The FTIR spectra of the electrolytes remain unchanged after 50 cycles. This clearly indicates that there is almost negligible side-reactions occurred during the battery reactions in the 50 cycles.

Figure R9 | Characterization of IL-TEMPO after cycling. a. UV spectra comparison of the pristine electrolyte and the electrolytes after the first cycle (discharge-charge) and 50 cycles. **b.** The corresponding FTIR spectra of the electrolytes.

The UV and FTIR spectra and corresponding description have been added as Figure S23 on Page 28 in the revised Supplementary Information.

The samples were prepared by soaking the electrolyte-adsorbed glass fibre separator in 3 ml DEGDME solvent, and the solutions were directly used for UV tests. The UV spectrum of the electrolyte after the first cycle shows that the concentration of IL-TEMPO slightly reduces, due to the incorporation of small amount of IL-TEMPO during the formation of SEI on the lithium anode. However, the concentration of IL-TEMPO stabilizes during the following cycles. The concentration after 50 cycles is almost the same as the one obtained after the first cycle. There is no additional peaks observed in the UV spectra. Furthermore, the FTIR spectra of the electrolytes remain unchanged after 50 cycles. This clearly indicates that there is almost negligible side-reactions occurred during the battery reactions in the 50 cycles.

Comment 7: *It is important to quantify the amount of Li_2O_2 during discharge. Can authors add the yield measurements (DOI: 10.1021/jz401659f) to quantify the amount under O_2 , 50% IL-TEMPO and air systems?*

Response 7: We would like to thank the reviewer for the valuable suggestion. Following the reviewer's comment, we have performed titration on the discharged electrodes that are obtained in the atmosphere of O_2 , and air using different electrolyte systems. The results are shown below as Table R3. As shown in the table, the discharge products of the cells are all dominated by Li_2O_2 , both in O_2 and air atmosphere. The cells with 1 % IL-TEMPO electrolyte operated in O_2 atmosphere show a maximum Li_2O_2 yield of 94.4 % of Li_2O_2 , indicating the exceptional catalytic activity and stability of IL-TEMPO during the operation of Li- O_2 batteries. The Li_2O_2 yield slightly decrease to around 85 %, owing to the existence of H_2O and CO_2 in the air atmosphere which could inevitably lead to small amount of side-reactions. Nevertheless, the exceptional capability of IL-TEMPO could still ensure the formation of Li_2O_2 as the majority of discharge products in the air atmosphere. The cells with 50 % IL-TEMPO electrolyte give a wide range of Li_2O_2 from 87% to 40%, probably due to the formation of large quantity of IL-TEMPO- O_2 intermediates which remain in the electrolyte (this is consistent with the *in-situ* DEMS results in Figure 32). However, it can be still counted as evidence that such high content of IL-TEMPO could still result in Li_2O_2 as the main discharge product.

Table R3 Discharge Li₂O₂ percent yield (Y_{Li₂O₂}) for the cells under O₂ and air atmosphere

Electrolyte	Atmosphere	Capacity (mAh)	Li ₂ O ₂ (μmol)	Y _{Li₂O₂} (%)
1 % IL-TEMPO	O ₂	2.08±0.02	36±1	92.7±1.7
	Air	2.08±0.02	32±2	82.6±4.5
50 % IL-TEMPO	O ₂	1.10±0.02	13±5	63.2±24.1
	Air	1.10±0.02	14±3	67.5±13.4

The above table has been added as Table S2 on Page 46 in the revised Supplementary Information, and the associate description of the experimental procedure has also been added on Page 5 in the experimental section of revised Supplementary Information. The recommended paper has been cited as Reference 52 in the revised Manuscript, and Reference 5 in the revised Supplementary Information.

52 McCloskey, B. D. et al. Combining accurate O₂ and Li₂O₂ assays to separate discharge and charge stability limitations in nonaqueous Li-O₂ batteries. *J. Phys. Chem. Lett.* **4**, 2989-2993 (2013).

Reviewer #2 (Remarks to the Author):

I have carefully reviewed the paper by Zhang et al., concerning the development of IL-TEMPO as a potential redox mediator and also as a designer electrolyte solvent for Li-O₂ and Li-air batteries. The area of research is exciting and the designer electrolytes for Li-O₂ have been called for by several researchers in the field as conventional solvents seem to not be stable within the electrochemical operating conditions of Li-O₂ and Li-air batteries. In fact, this seems to be true for other alkali metal-oxygen chemistries as well. Therefore, I think that this paper warrants publication in Nature Communications. However, I have the following serious concerns and therefore advise that the paper be considered for publication after the following are fully addressed:

Comment 1: *The authors in Page 6 line 113-115 suggest that the shift in the N-O reduction potential is possibly due to SEI formation on the Lithium Anode during electrochemical CV measurements. I suggest that Impedance data on the Li Anode to corroborate this hypothesis be provided.*

Response 1: We would like to thank the reviewer for the valuable comment. Following the reviewer's suggestion, we have measured a.c. impedance data to further demonstrate the SEI

formation on the lithium anode. The a.c. impedance spectra were measured after each cycle of the CV test. The results are shown in Figure R10. The impedance of the cell slightly increases after the first cycle, and stabilizes in the following scans. This indicates that SEI layer is formed during first cycle, which remains stabilized. The incorporation of IL-TEMPO molecules in the SEI layer induces a relatively low interface resistance because the existence of IL-TEMPO can efficiently enhance the ionic transport through the SEI layer. Combining with the CV results, it can be concluded that the shift of the first pair of reduction peaks is due to the formation of SEI layer on the lithium metal in the first cycle.

Figure R10 | The impedance spectra of the cell after each cycle during the CV test. a. The CV tests and **b.** the corresponding impedance spectra of the cell. The lithium metal is not pre-treated to form SEI layer in advance. The scan rate was 0.5 mV s⁻¹.

The a.c. impedance spectra and the associated descriptions have been added as Figure S3 on Page 8 in the revised Supplementary Information.

The potential for the reduction of the N-O radical undergoes a positive shift after the first cycle as shown in Figure S3a, which could be due to the formation of a solid electrolyte interface (SEI) on the surface of the lithium anode. This has been proven by monitoring the a.c. impedance spectra change after each cycles in Figure S3b. Meanwhile, the CV profiles in the second and third cycles (Figure S3a) match perfectly with the CV curves obtained using pre-treated lithium anode (Figure 1b), further confirming the SEI formation during the CV tests.

Comment 2: A major concern in the paper is the redox potentials for IL-Tempo and Li-O₂ cells. The two prominent redox peaks in the CV for IL-TEMPO are at 3 V and 3.6V. A nearly flat

potential during a galvanostatic charge step of Li-O₂ cells being around 3.5-3.6 V is reasonable assuming redox mediation by IL-TEMPO. However, the galvanostatic discharge potential for Li-O₂ cells is ~ 2.7 V and redox mediation by IL-TEMPO during discharge is not expected. The authors should clearly explain this apparent discrepancy.

Response 2: We would like to thank the reviewer for the valuable comment. The CV curves of IL-TEMPO indicate that there are two distinct pairs of peaks corresponding to the reduction (around 3 V) and oxidation (3.5-3.8 V) of the N-O radical group in IL-TEMPO. The oxidation potential of the N-O radical group in IL-TEMPO is higher than the theoretical decomposition potential of Li₂O₂ (2.96 V), yet lower than the actual polarized charging potential (usually higher than 4.2 V). Therefore, the incorporation of IL-TEMPO in the electrolyte can significantly reduce the charge over-potential by the direct reaction between the oxidized IL-TEMPO and Li₂O₂. Therefore, IL-TEMPO mainly functions as a redox mediator during the charging process. Similarly, the reduction potential of IL-TEMPO should be very close to the theoretical formation potential of Li₂O₂ (2.96 V) which is higher than the actual potential of discharge (usually lower than 2.75 V) in order to allow the reduced IL-TEMPO to participate in the ORR process (*Nat. Mater.* 15, 918, 2016; *Adv. Mater.* 30, 1705571, 2018). A reduction potential of ~ 3.0 V ensures that IL-TEMPO molecules can be reduced before the formation of Li₂O₂ occurs during the discharge process. As a result, the reduced IL-TEMPO could efficiently interact with the oxygen species when the ORR starts to proceed (as shown in the demonstration experiment with chemically reduced IL-TEMPO in Figure S9). The interaction allows the IL-TEMPO functioning as a redox shuttle, which would facilitate the formation of Li₂O₂ and leave IL-TEMPO back to its original radical form. Since the ORR potential (~2.7 V) is lower than the reduction potential (~3.0 V), the IL-TEMPO molecules can be continuously reduced, leading to the enhancement of the discharge process. Therefore, the discharge capacity is significantly improved, which is consistent with discharge-charge curves in Figure 1c and CV curves in O₂ atmosphere in Figure S5-S6.

The explanation has been added on Page 6 in the revised Manuscript for better addressing the statement.

In order to enhance both the discharge and charge processes during the operation of Li-O₂ batteries, the reduction potential of the mediator molecules should be very close to the theoretical formation potential of Li₂O₂ and higher than the actual discharge plateau (~ 2.7 V),

while the oxidation potential should be higher than the theoretical decomposition potential of Li_2O_2 yet lower than the actual charge plateau ($\sim 4.2 \text{ V}$)^{20,21}. The potentials of the reversible peaks (3.0 V and 3.75 V) of IL-TEMPO perfectly match the above required potential windows where the non-aqueous oxygen reduction reaction (ORR) and oxygen evolution reaction (OER) are enhanced in Li-O₂ batteries (theoretical potential 2.96 V, Figure 1b).

Comment 3: *The authors have performed CVs in an inert Argon atmosphere to show redox behavior of IL-TEMPO. However, a more pertinent and useful measurement is to perform CVs using inert electrodes in the presence of oxygen. This is necessary to understand and irreversibility inherent to IL-TEMPO in the presence of oxygen which is the more realistic operating condition. Unless this experiment, the authors cannot conclude beyond doubt that the contribution of IL-TEMPO self-redox to capacity is negligible.*

Response 3: We would like to thank the reviewer for this valuable suggestion. The CV results in an argon atmosphere are used to reveal the redox behaviour of IL-TEMPO without the interference of oxygen reduction reaction. We have also performed CV tests in an oxygen atmosphere with CNT electrodes further to illustrate the facilitation of ORR and OER processes by IL-TEMPO. Following the reviewer's suggestion, in order to further understand the redox behaviour of IL-TEMPO, we performed the CV tests in the oxygen atmosphere with only a stainless steel mesh current collector (no active carbon catalyst) as the inert electrode. The results are shown below as Figure R11. Due to the inert nature and relatively low surface area of the stainless steel mesh, the intensities of ORR and OER in the DEGDME electrolyte have been significantly reduced, comparing to the results with CNT electrode. However, the CV curve of the cell with IL-TEMPO electrolyte shows the distinctive ORR and OER peaks, revealing its extraordinary catalytic capability. Furthermore, the intensities of the peaks corresponding to the oxidation and reduction of IL-TEMPO are visible, yet faint comparing to the peaks related to ORR and OER. This demonstrates that the contribution of IL-TEMPO self-redox to capacity is negligible. The possible reason for the slight differences between Figure S5 and S6 could be that the stainless steel current collector does not have the same porous structure as the carbon paper. Therefore, the IL-TEMPO molecules could easily diffuse away from the electrode, resulting in the slight decrease of the reversible peaks corresponding to the reduction/oxidation of IL-TEMPO.

Figure R11 | The CV curves of the Li-O₂ cells with stainless steel mesh current collector as cathode in O₂ atmosphere. The scan rate is 0.5 mV s⁻¹.

The above CV curves have been added as Figure S6 on Page 11 in the revised Supplementary Information.

Comment 4: *The very small discharge capacity of Li-O₂ cells in a DEGDME electrolyte without IL-TEMPO is not expected. The authors must compare their ultimate areal and gravimetric capacities with other published work to show that the capacities obtained in their measurements are not much lower than published work.*

Response 4: We would like to thank the reviewer for this valuable comment. The carbon paper we used is directly purchased from Quintech, which shows a BET surface area below 1 m² g⁻¹ and the roughness factor (total surface area/areal area) is 90. Such low surface area would result in a very small discharge capacity when used in Li-O₂ cells, which in turn makes it perfect to be used as the reference cathode to identify the promotion capability of the redox mediator for discharge process. The discharge capacities in this manuscript are calculated based on the area of the carbon paper electrode (0.785 cm²). Following the reviewer's comment, the obtained area capacity and actual capacity for the carbon paper electrode in oxygen atmosphere are summarized below as Table R4. The capacities calculated from relevant references with the same carbon paper are also included for comparison. As shown in the Table, the capacities of the carbon paper obtained in this work are similar to the data reported in previous publication.

Table R4 | The comparison of the capacities when carbon paper (H2315, Quintech) is used.

	Areal Capacity (mA cm ⁻²)	Real Capacity (mA)
This work	0.095	0.075
Nat. Mater. 15, 882, 2016	0.122	0.019
Nat. Energy 2, 17118, 2017	0.100	0.016
Nat. Mater. 12, 228, 2013	0.055	0.062

The information of the carbon paper has been added to the experimental part on Page 3 in the revised Supplementary Information.

Carbon paper electrodes were prepared by stacking three layers of pre-cut carbon papers (10 mm diameter, H2315, Quintech).

Comment 5: *X-ray diffractograms presented in this manuscript are in general not rigorously indexed. I suggest that all XRD patterns be indexed according to standard JCPDS patterns and ensure that the peak positions and intensity ratios are in agreement with JCPDS standards. This is very much necessary as some of the high intensity peaks for LiOH and Li₂O₂ are very close in 2-Theta values.*

Response 5: We would like to thank the reviewer for the valuable suggestion. Following the reviewer's suggestion, we have indexed all XRD patterns in the Supplementary Information (Figure S7, S17a, S30, and S35) with the standard Li₂O₂ patterns, and displayed below in Figure R12. The results all confirm that the discharge products are all crystalline Li₂O₂ (except for the one in 50 % IL-TEMPO, which is amorphous Li₂O₂, confirmed by the FTIR spectrum in Figure S31 and DEMS result in Figure S32).

Figure R12 | The XRD patterns of the electrodes with standard JCPDS index. The electrodes are obtained in Li-O₂ cells with **a.** carbon paper electrode in 1 % IL-TEMPO electrolyte in O₂, **b.** CNT electrode in 1 % IL-TEMPO electrolyte in O₂, **c.** CNT electrode in 1 % IL-TEMPO electrolyte in air, **d.** CNT electrode in 50 % IL-TEMPO electrolyte in O₂.

Comment 6: According to Figure 3, the DEMS data shows that the Li-O₂ cells with IL-TEMPO evolve more oxygen than they consume during discharge. Isn't this a serious issue? What is the error bar on these measurements?

Response 6: We would like to thank the reviewer for the valuable comment. We did the integration for the O₂ consumption and evolution curves, in order to compare the ratio between the amounts of O₂ consumed during discharge and released during charge. The results are shown in Figure R13 and Table R5. The calculated results show the O₂ evolution/consumption ratio is slightly smaller than 1, indicating that the released amount of O₂ is slightly lower than the consumed amount. The DEMS tests may induce about 5 % error, which is usually caused

by the inevitable trace amount of side-reactions during the discharge process. It can be also induced by the release of trace amount of remaining O₂ which may be originated from the leftover in the electrolyte after the degassing process. By comparing the results shown in Table S3 in the revised Supplementary Information, the values lie in the reasonable range when the 5 % error is taken into account.

Figure R13 | *In-situ* DEMS analysis of the gas consumption and evolution during Li-O₂ cell operation. The electrolyte used is 1 % IL-TEMPO electrolyte.

Table R5 | The O₂ consumption/evolution amounts obtained by integration of Figure R13.

	O ₂ consumption	O ₂ evolution	Evolution/Consumption
Integration	1.530	1.514	98.95%

The error bar of the DEMS results has been included in Table S3 on Page S46 in the revised Supplementary Information.

The error bar during this test is considered as 5 %.

Comment 7: *The authors have used a capacity of 0.25 mAh/cm² for cycling measurements. This is more than 15 times smaller than the ultimate capacity of the cells. A common standard in battery research is to cycle these cells to at least 80% of their full capacity. While cyclability at a depth of discharge of at least 80% is not necessary publishing this work, such measurements reveal the limitations of the chemistry and provide inspiration for future work.*

Response 7: We would like to thank the reviewer for the valuable suggestion. The reviewer

has provided a very meaningful insight which we will surely pay attention to in the future work. In the meantime, we performed additional cycling measurements with extended capacity from 1 mA cm^{-2} to 2 mA cm^{-2} , in order to further illustrate the capability of IL-TEMPO to facilitate the cycling performance of Li-O₂ batteries. All the results in Figure R14 indicate that the addition of IL-TEMPO in the electrolyte could significantly enhance the cycle life of Li-O₂ batteries.

Figure R14 | The discharge-charge profiles of Li-O₂ batteries with 1 % IL-TEMPO electrolyte. a. The capacity was restricted to 1 mAh cm^{-2} . The current density was 0.1 mA cm^{-2} . **b.** The capacity was restricted to 2 mAh cm^{-2} . The current density was 0.2 mA cm^{-2} .

The above cycling data have been added as Figure S22 on Page 27 in the revised Supplementary Information. The corresponding description has been added in the main text on Page 10 in the revised Manuscript.

Therefore, there should be a negligible amount of side-products formed (confirmed by titration of Li₂O₂ in Table S2), which leads to the exceptional cycling performance as shown in Figure S21-22.

Comment 8: *The authors seem to have incorrect units in certain figures and captions. For example Fig. S24 and caption in Fig. 3. gravimetric normalization is used in these two instances while the rest of the paper uses areal normalization.*

Response 8: We would like to thank the reviewer for this comment. We have checked and revised all the incorrect units in the captions of Figure 3 and Figure S32 (original Figure S24).

The incorrect units in Table S3 are also revised.

Comment 9: Page 15 Lines 312 - 321: The authors suggest that the absence of Li_2O_2 peaks in XRD is indicative of amorphous Li_2O_2 . However, there is no evidence for this assumption. For example, the authors could perform peroxide titrations or Raman/FTIR measurements to confirm the presence of Li_2O_2 . Without such evidence, it will be hard to convince the readers of the presence of Li_2O_2

Response 9: We would like to thank the reviewer for the valuable suggestion. We have followed the reviewer's suggestion and performed the suggested measurement. In order to further confirm the formation of amorphous Li_2O_2 during the operation of Li-O₂ batteries with 50 % IL-TEMPO, we performed FTIR (Figure R15) and titration (Table R6) on the discharged electrode. As shown in the FTIR spectra, the characteristic peaks of Li_2O_2 in the range of 400 to 600 cm^{-1} can be observed, confirming the discharge product is dominated by Li_2O_2 . Furthermore, the titration result and the *in-situ* DEMS result (Figure S32) all provide solid evidence that Li_2O_2 is formed during the discharge process. The relatively lower yield of Li_2O_2 obtained by the titration method is due to the formation of large quantity of IL-TEMPO-O₂ intermediate existing in the electrolyte (which is also consistent with the *in-situ* DEMS result and demonstration experiment in Figure S9). However, there is no characteristic peaks spotted in the XRD spectrum. Therefore, it can be concluded that high content (50 %) of IL-TEMPO could lead to the formation of amorphous Li_2O_2 .

Figure R15 | The FTIR spectra of the electrode before and after cycles using the 50 % IL-TEMPO electrolyte.

Table R6 Discharge Li_2O_2 percent yield ($Y_{\text{Li}_2\text{O}_2}$) for the cells with 50% IL-TEMPO electrolyte under O_2 atmosphere

Electrolyte	Atmosphere	Capacity (mAh)	Li_2O_2 (μmol)	$Y_{\text{Li}_2\text{O}_2}$ (%)
50 % IL-TEMPO	O_2	1.10 ± 0.02	13 ± 5	63.2 ± 24.1

The FTIR spectra and titration results have been added as Figure S31 on Page 36 and Table S2 on Page 46 in the revised Supplementary Information. The description has also been added in the revised Manuscript at Page 15.

It is further confirmed by the FTIR spectra (Figure S31) and titration results (Table S2) of the electrode after discharge.

Comment 10: *Page 16 Line 346 - 348: There is no evidence anywhere in the paper to suggest that these batteries could work at elevated temperatures of up to 70 C. I suggest that the authors delete this sentence from the concluding paragraph.*

Response 10: We would like to thank the reviewer for this suggestion. The sentence from the concluding paragraph on Page 16 has been deleted accordingly.

Comment 11: *Figs 5d, S24 and S27 are contradicting Figures S15. While the authors show that Li_2CO_3 is not oxidized by IL-TEMPO, the experiments in Li-Air cells where carbonate formation is expected suggest that there is negligible difference between the galvanostatic charge curves for Li- O_2 and Li-air cells. This is evidence that Li_2CO_3 is likely oxidizing at around 3.6 V. Also, Fig. S24 clearly shows a small but observable increase in CO_2 evolution. The authors need to address this discrepancy.*

Response 11: We would like to thank the reviewer for the comment. Our demonstration experiment in the Figure S20 (original Figure S15) uses commercial Li_2CO_3 and Li_2O_2 to distinguish the capability of IL-TEMPO to catalyse the Li_2O_2 and Li_2CO_3 . The results show a charging plateau around 3.7 V for Li_2O_2 -loaded electrode while 4.6 V for Li_2CO_3 -loaded electrode. This provides a general conclusion that IL-TEMPO have excellent capability to decompose Li_2O_2 while have negligible impact on the crystalline Li_2CO_3 . However, due to the

complexity of the air battery system, the existence forms of the possible side-products are not limited to Li_2CO_3 . Various lithium organic carbonates are also reported as intermediate side-products formed during the operation of Li-O₂ batteries, which are easier to be decomposed than the crystalline Li_2CO_3 (*J. Am. Chem. Soc.* 133, 8040, 2011; *J. Phys. Chem. C* 116, 19724, 2012). Even in the air system, the combination of ether-based electrolyte and hydrophobic IL-TEMPO with excellent catalytic activity could efficiently reduce the possibility to form large amount of high crystalline Li_2CO_3 (*J. Am. Chem. Soc.* 135, 9733, 2013). Furthermore, the electrochemically formed Li_2CO_3 during the battery operation may have lower crystallinity than the commercial available Li_2CO_3 , thus higher activity towards decomposition. Therefore, it may be possible that IL-TEMPO could catalyse the decomposition of these compounds if existed. However, our characterization results of XRD and FTIR all indicate that there are minimum products other than Li_2O_2 formed during the battery operation. Therefore, the small amount of CO_2 evolution during the charging process may originate from the trace amount of intermediate side-products or the decomposition of small amount electrolyte. But in general, the battery reactions are highly reversible owing to the excellent capability of IL-TEMPO towards the formation and decomposition of Li_2O_2 during discharge and charge processes.

Relevant sections on Page 10 in the revised Manuscript and on Page 25 in the revised Supplementary Information have been carefully revised to more precisely address this concept.

In the revised Manuscript: Furthermore, as proven by our designed experiment in Figure S20, IL-TEMPO is only fully capable of decomposing Li_2O_2 , but not Li_2CO_3 with high crystallinity.
In the revised Supplementary Information: The results indicate that the IL-TEMPO could catalyse the decomposition of Li_2O_2 other than crystalline Li_2CO_3 .

Comment 12: *Finally, none of the galvanostatic charge curves polarize similar to the polarization observed in Fig. S15. A potential polarization is expected at the end of charge for any electrochemical devices. The authors must address such as polarization is not observed in all of their limited capacity cycling data.*

Response 12: We would like to thank the reviewer for the comment. It has been demonstrated that IL-TEMPO functions as redox mediator during the charge process, which can efficiently catalyse the decomposition of Li_2O_2 and lead to a dramatically reduced charge over-potential. Furthermore, it has also been demonstrated that the addition of IL-TEMPO could significantly

enhance the stability of Li-O₂ batteries and reduce the risk of side-reactions by facilitating the discharge and charge processes, and forming protective SEI layer on the lithium metal. Therefore, the round-trip efficiency can be significantly improved, explaining such low potential polarization during the entire charge process. Furthermore, to further identify the chemistry during the charge process, we performed an experiment by overcharging the Li-O₂ in the first cycle. As shown in Figure R16a, the charge capacity is slightly higher until an expected potential polarization occurs. This is because IL-TEMPO functions as a redox mediator during the charge process, and the accumulation of oxidizing IL-TEMPO may contribute a little to the charge capacity when all the discharge products (Li₂O₂) are consumed. The overcharged capacity is consistent with the self-redox capacity shown in Figure S15. Consequently, additional discharge capacity occurs before the formation of Li₂O₂ which attributes to the reduction of the oxidized IL-TEMPO. The significant potential polarization in Figure S20 (original Figure S15) is also based on the same overcharge principle. However, during the normal operation of Li-O₂ batteries in this work (comparison Figure R16b), the discharge and charge capacities are kept the same. Owing to such good capability in enhancing the stability of Li-O₂ batteries, no significant increase of charge over-potential is observed at the end of the charging process and no additional discharge capacity of reducing IL-TEMPO is observed at the beginning of the following discharge process before the formation of Li₂O₂, indicating that the discharge and charge capacities are fully contributed by the formation and decomposition of Li₂O₂.

Figure R16 | The discharge-charge profiles of Li-O₂ cell with 1 % IL-TEMPO electrolyte.
a. The cell is overcharged to 0.35 mAh cm⁻² after discharge to 0.25 mAh cm⁻². **b.** The cell is normal-charged to 0.25 mAh cm⁻² after discharge to 0.25 mAh cm⁻². The current density was 0.1 mA cm⁻².

The above discharge-overcharge profiles has been added as Figure S19 on Page 24 in the revised Supplementary Information. The corresponding description has been added on Page 24 in the revised Supplementary Information.

The charge capacity is slightly higher than the discharge capacity until an expected increase of over-potential is observed. The additional capacity is contributed by the self-redox of the IL-TEMPO which has similar value as in Figure S15a. The consequent discharge curve shows additional capacity before the beginning of the Li_2O_2 formation, which is directly related to the reduction of the accumulated oxidized IL-TEMPO during the overcharge process. The distinctively different discharge curves in the non-overcharge curves proves that the contribution of the self-redox of IL-TEMPO is negligible in the normal operation of Li-O₂ batteries.

Reviewers' comments:

Reviewer #1 (Remarks to the Author):

The authors answered my questions very well. The data now is quite convincing. It is quite nice work and I am happy to recommend publication of this work.

Reviewer #2 (Remarks to the Author):

After going through the revisions submitted, I feel that the manuscript has been significantly improved. However, the following still need addressing before the paper can be deemed ready for publication.

1) The Data shown in Table R5 contradicts with e/O₂ numbers in Figure 3e and 3f. If the calculations shown in Table R5 are correct, the e/O₂ during charge should be greater than e/O₂ during discharge (assuming that the cells were discharged/charged to the same capacity). This is not the case.

2) Please include an error bar for the DEMS data in Fig 3 or at the least mention error in e/O₂ calculation in the fig caption.

3) Cycling measurements with discharge/charge set to 1 mAh/cm² or 2 mAh/cm² clearly show that the cells tends to degrade much sooner. This is clear evidence that there are potential parasitic reactions/products which are below the measurement thresholds of several characterisation techniques used in this work. The authors should clearly mention the possibilities for such parasitic reactions or at the least discuss this as a potential outstanding issue.

Reviewer #2 (Remarks to the Author):

After going through the revisions submitted, I feel that the manuscript has been significantly improved. However, the following still need addressing before the paper can be deemed ready for publication.

Comment 1: The Data shown in Table R5 contradicts with e/O_2 numbers in Figure 3e and 3f. If the calculations shown in Table R5 are correct, the e/O_2 during charge should be greater than e/O_2 during discharge (assuming that the cells were discharged/charged to the same capacity). This is not the case.

Response 1: We would like to thank the reviewer for the valuable comment. According to the reviewer's suggestion, we re-examined the results in Table R5, Table S3, and Figure 3e-f to further confirm the consistency. After the comparison, we realize that the calculated results in Table R5 are incorrect which should be discarded, and the results in Table S3 are correct and reliable. The amount of released oxygen is slightly higher than the amount of consumed oxygen, which is caused by the error induced during DEMS tests. Due to the use of different gas atmosphere (stated in the experimental section), the errors are actually different during discharge and charge processes. The error induced by DEMS during the discharge process is 5 % when a mixture of Ar/O₂ (1:4) is used. During the charge process, only pure Ar is used as the carrier gas, which makes the O₂ signal obtained more accurate than the data obtained during the discharge process. A slightly smaller error of 3 % is induced. Therefore, the results in Table S3 are reasonable when the errors are considered. In order to clearly address this issue, we have added the error value in Table S3 on Page 45 in the revised Supplementary Information.

Table S3 | DEMS results of Li-O₂ batteries with different electrolytes

	Charge passed ^a		O ₂ quantity		e^-/O_2		CO ₂ quantity	
	D ^b	R ^b	C ^b	E ^b	D ^b	R ^b	C ^b	E ^b
DEGDME electrolyte	18.65	17.89	9.01±0.45	8.43±0.25	2.07	2.12	-	0.464
IL-TEMPO electrolyte (1%)	18.64	18.64	9.18±0.45	9.49±0.28	2.03	1.96	-	0.134

The error during discharge is considered as 5 %, and during charge is considered as 3 %.

Comment 2: Please include an error bar for the DEMS data in Fig 3 or at the least mention error in e/O_2 calculation in the fig caption.

Response 2: We would like to thank the reviewer for the valuable suggestion. The description of the error has been added in the fig caption on Page 19 in the revised Manuscript.

The error of the DEMS data obtained during discharge is 5 %, and during charge is 3 %.

Comment 3: Cycling measurements with discharge/charge set to 1 mAh/cm² or 2 mAh/cm² clearly show that the cells tends to degrade much sooner. This is clear evidence that there are potential parasitic reactions/products which are below the measurement thresholds of several characterisation techniques used in this work. The authors should clearly mention the possibilities for such parasitic reactions or at the least discuss this as a potential outstanding issue.

Response 3: We would like to thank the reviewer for the valuable suggestion. We have followed the reviewer's suggestion and added the discussion of the potential outstanding issue in the Manuscript and Supplementary Information. The side-reactions during the operation of Li-O₂ batteries have been significantly suppressed by the use of the versatile IL-TEMPO, which could facilitate both discharge and charge process and stabilize the lithium anode. However, the extension of the capacity limitation during cell cycling also extended the exposure time of the discharge product, Li₂O₂, to the glyme electrolyte. Since Li₂O₂ has been demonstrated to be unstable when kept in the atmosphere of glyme for a long period of time, this would certainly lead to the formation of more by-products, which could negatively affect the cycling performance of Li-O₂ cells (*J. Phys. Chem. Lett.*, 3, 3043-3047, 2012; *J. Phys. Chem. Lett.*, 2, 1161-1166, 2011; *Nat. Energy*, 2, 17036, 2017; *Chem. Mater.*, 25, 77-84, 2012; *J. Phys. Chem. Lett.*, 4, 2989-2993, 2013; *Angew. Chem. Int. Ed.*, 50, 8609-8613, 2011). The extension of the capacity could also lead to the formation of large quantity of Li₂O₂, which could also induce more side-reactions due to more direct contact between carbon and Li₂O₂ (*J. Phys. Chem. Lett.*, 3, 997-1001, 2012; *J. Am. Chem. Soc.*, 135, 494-500,

2013; *Nat. Energy*, 2, 17118, 2017; *Adv. Energy Mater.*, 8, 1702661, 2018). Furthermore, deep discharge may result in the blockage or even expansion of the electrode which would lead to the loss of contact between the CNTs and Li_2O_2 in the electrode (*Angew. Chem. Int. Ed.*, 47, 4521-4524, 2008). This may further reflected in the subsequent recharge. Nevertheless, the Li- O_2 cells could still managed to last reasonable cycle lives when the capacity was set to 1 mAh cm^{-2} or 2 mAh cm^{-2} , which clearly indicates the exceptional capability of IL-TEMPO to increase the battery cycling performance.

We have followed the reviewer's suggestion and added the necessary description of parasitic reactions on Page 7 in the revised Manuscript and Page 26 in the revised Supplementary Information.

Manuscript: It is worth noting that the extension of discharge/charge capacity does not significantly deteriorate the cycling performance, even with the risk of generating more by-products due to the parasitic reactions between the discharge product Li_2O_2 and ether-based electrolyte.

Supplementary Information: The extension of the capacity could potentially increase the possibility of side-reactions between the metastable Li_2O_2 and the DEGDME electrolyte/carbon electrode due to the extended exposing time. Furthermore, the large quantity of discharge products may cause the expansion of CNT electrode, which could lead to the loss of contact between the electrode materials, thus negatively affected the cycle life of Li- O_2 cells. Nevertheless, the Li- O_2 cells still maintain exceptional cycling performance with the capacity limitations extended to 1 mAh cm^{-2} and 2 mAh cm^{-2} , indicating the exceptional capability of IL-TEMPO to increase the cycling performance of Li- O_2 batteries.